# Hup-Type Hydrogenases of Purple Bacteria: Homology Modeling and Computational Assessment of Biotechnological Potential

**DOI:** 10.3390/ijms21010366

**Published:** 2020-01-06

**Authors:** Azat Vadimovich Abdullatypov

**Affiliations:** Institute of Basic Biological Problems of the Russian Academy of Sciences—A Separate Subdivision of PSCBR RAS (IBBP RAS), Institutskaya str., 2, Pushchino, 142290 Moscow, Russia; azatik888@yandex.ru; Tel.: +7-(4967)-73-27-91

**Keywords:** hydrogenases, hydrogen fuel cells, homology modeling, FeS clusters, transmembrane helices, molecular docking, molecular tunnels

## Abstract

Three-dimensional structures of six closely related hydrogenases from purple bacteria were modeled by combining the template-based and ab initio modeling approach. The results led to the conclusion that there should be a 4Fe3S cluster in the structure of these enzymes. Thus, these hydrogenases could draw interest for exploring their oxygen tolerance and practical applicability in hydrogen fuel cells. Analysis of the 4Fe3S cluster’s microenvironment showed intragroup heterogeneity. A possible function of the C-terminal part of the small subunit in membrane binding is discussed. Comparison of the built models with existing hydrogenases of the same subgroup (membrane-bound oxygen-tolerant hydrogenases) was carried out. Analysis of intramolecular interactions in the large subunits showed statistically reliable differences in the number of hydrophobic interactions and ionic interactions. Molecular tunnels were mapped in the models and compared with structures from the PDB. Protein–protein docking showed that these enzymes could exchange electrons in an oligomeric state, which is important for oxygen-tolerant hydrogenases. Molecular docking with model electrode compounds showed mostly the same results as with hydrogenases from *E. coli*, *H. marinus*, *R. eutropha*, and *S. enterica*; some interesting results were shown in case of HupSL from *Rba. sphaeroides* and *Rvi. gelatinosus*.

## 1. Introduction

Purple bacteria are widespread organisms possessing tremendous potential for practical application. They are producers of many valuable compounds, such as photosynthetic pigments (carotenoids [1] and bacteriochlorophyll [2]), storage compounds (polyhydroxyalkanoates [3]), phytohormones [4], and molecular hydrogen [5]. Another promising application of purple bacteria is their use for enzyme production. For example, there is already one enzyme available on the market, namely 3-hydroxybutyrate-dehydrogenase, from *Rhodobacter sphaeroides* [6], which is used for analytical quantifications of 3-hydroxybutyrate and acetoacetate; besides these enzymes, purple non-sulfur and sulfur bacteria possess hydrogenase enzymes. This group of enzymes has been studied for almost nine decades since its discovery by Stephenson and Stickland [7].

For example, Hyd-type hydrogenases from purple sulfur bacteria (*Chromatium vinosum* (*Allochromatium vinosum*) and *Thiocapsa roseopersicina*) have been used in hydrogen electrodes, which could act as components of fuel cells and hydrogen biosensors [8,9]. These enzymes are relatively oxygen-tolerant and thermostable. Nevertheless, they are reversibly inhibited in presence of low oxygen concentrations, although other studies demonstrated their oxygen tolerance in some experimental conditions [10,11,12]. Anyway, they lack clear determinants of stability vs. oxygen as described for other groups of hydrogenases. Since a fuel cell should be suitable for storage in air and operation in hydrogen, meaning that there could be steps of exposition of hydrogenase electrodes to hydrogen-oxygen mixture, this gives rise to some obstacles for the wide application of these enzymes in hydrogen electrodes.

On the other hand, there are hydrogenases demonstrating higher levels of oxygen tolerance than HydSL enzymes from *Thiocapsa roseopersicina* and *Chromatium vinosum*. Such enzymes are hydrogenases from *Hydrogenovibrio marinus*, *Ralstonia eutropha*, *Salmonella enterica*, and hydrogenase-1 from *Escherichia coli*. According to X-ray crystallographic and electrochemical studies, these hydrogenases possess a 4Fe-3S cluster [13,14,15,16]. The mechanism providing their oxygen tolerance includes oligomerization: They are capable of reducing the oxidized active site by electrons from the FeS clusters of an adjacent hydrogenase molecule from a dimer of heterodimers [17].

Purple bacteria are known to possess specific hydrogen-sensing hydrogenases which were demonstrated to be absolutely insensitive to the presence of molecular oxygen or air in hydrogen gas, as well as to acetylene [18]. However, hydrogen-sensing hydrogenases are the least active enzymes of this class, oxidizing only 5–20 μmol H_2_ per mg protein per minute, which is apparently due to the narrowness of gas access channels, thus low activity is an inevitable price to be paid for absolute oxygen tolerance [19].

However, there is another group of hydrogenases in purple bacteria, namely HupSL-hydrogenases [20,21,22,23,24,25]. They haven’t drawn much attention as in vitro catalysts for a while, but recently discovered 4Fe3S clusters led to hypothesis that they should be treated as possible oxygen-tolerant hydrogen oxidation catalysts like the mentioned membrane-bound hydrogenases, which are their close homologues. Hence, their structural modeling appeared to be interesting in the prospect of their biotechnological application. 

The structure of HupSL hydrogenase from purple sulfur bacterium *Thiocapsa roseopersicina* was modeled long ago by Szilagyi and colleagues [26]; moreover, it was studied experimentally. It was considered as unstable and oxygen-sensitive, but the experimental conditions did not exclude proteolysis factors: The protein was isolated only as a fraction of DEAE-chromatogram of crude cell extract, which lost almost all its activity at 4 °C under air [27], and it was proposed that the reason for its instability was a low number of intersubunit ionic pairs [26].

Hydrogenase from *Rhodobacter capsulatus* was studied by a number of research teams. It was shown to participate in the recycling of hydrogen produced in the nitrogenase reaction. Among the in vitro studies, work of Magnani and colleagues should be noted. They suggested the possible existence of two binding sites for electron acceptors (methylene blue and benzyl viologen), and diphenylene iodonium was shown to be an inhibitor of electron transfer to both methylene blue and benzyl viologen, but the inhibition was complete only in the second case [28]. Such multiplicity of electron acceptor binding sites should be taken into account when designing electrochemical devices based on such enzymes.

*Rhodopseudomonas palustris* CGA009 was shown to be a HupSL-defective strain that was uncapable of growing on H_2_ as a sole electron donor. Genome sequencing revealed that it is caused by a defective HupUV-sensing hydrogenase, particularly, by a frame-shift mutation in the HupV gene. However, other *Rps. palustris* strains have a functional HupV gene and the corresponding ability to utilize hydrogen under nitrogen-fixing conditions provided by HupSL hydrogenase [23].

*Rubrivivax gelatinosus* was subjected to deletion of HupSL genes in order to increase hydrogen yield. This work by Wawrousek and colleagues mentioned presence of supernumerary cysteines in this enzyme, drawing interest to it from the point of view of oxygen tolerance [29]; the work by Manness and co-workers demonstrated presence of oxygen-tolerant hydrogenase linked to CO metabolism of this bacterium [30].

*Rhodospirillum rubrum* possesses several metabolically important hydrogenases; one of them is membrane-bound HupSL-hydrogenase. It was studied by Adams and Hall [31] and by Kakuno and colleagues [32]; its ability to be released into cultural medium and to tolerate high salt concentration is intriguing and promising.

The choice of electrode materials for hydrogenase electrodes is still a great challenge having various solutions. First of all, there are nonmodified carbon materials (carbon felt, carbon black, carbon fiber, carbon nanotubes, and pyrolytic graphite). As pyrolysis-derived carbon materials have nonzero level of oxidation, graphene oxide could be used as a model of graphite electrode plane. It is also a candidate for manufacturing bioelectocatalysts itself, and it was used in recent docking study with glucose oxidase by Sumaryada and co-workers [33].

Another approach is electropolymerization of redox-active dyes on the electrode surfaces. Voronin and colleagues tested several substances and found that electropolymerized neutral red-modified carbon felt was comparable to Vulcan XC72 carbon black by current densities of hydrogenase-based electrodes [34].

The goals of the current work were as follows: (1) to obtain homology models of hydrogenases from six purple bacteria, *Tca. roseopersicina*, *Rba. capsulatus*, *Rba. sphaeroides*, *Rps. palustris*, *Rvi. gelatinosus*, and *Rsp. rubrum*; (2) to study the microenvironment of the proximal FeS clusters; (3) to assess intrasubunit and intersubunit interactions between amino acid residues in these enzymes; (4) to assess the possibility of membrane binding of the C-terminal extensions of the small subunits; (5) to map intramolecular tunnels (possible pathways of hydrogen and oxygen delivery to the active site); (6) to assess the possibility of their oligomerization into complexes providing electron exchange between the small subunits; (7) to perform molecular docking of model electrode compounds, namely graphene oxide and neutral red oligomers, in order to estimate their applicability in fuel-cell electrodes compared to the membrane-bound hydrogenases from *E. coli*, *R. eutropha*, *H. marinus*, and *S. enterica*.

## 2. Results

The sequences of six hydrogenases modeled in the current study (HupSL enzymes from *Tca. roseopersicina*, *Rba. capsulatus*, *Rba. sphaeroides*, *Rps. palustris*, *Rvi. gelatinosus*, and *Rsp. rubrum*) are listed in Appendix B. All the models demonstrated common features of membrane-bound oxygen-tolerant hydrogenases. They are composed of two subunits, the large one bearing NiFe active site and magnesium ion, and the small one harboring three FeS clusters: 4Fe3S, 3Fe4S, 4Fe4S.

### 2.1. Modeling of Main (Aligned) Parts of the Hydrogenase Enzymes and Overview of Modeling Results

Several procedures were carried out to improve the quality of homology models built in MODELLER (see Methods section). One of the most valuable contributions to the overall quality of the models, according to z-DOPE assessment [35], was made by energy minimization on YASARA web server [36,37].

Despite the high homology levels of all the studied enzymes with X-ray crystallographic data, normalized DOPE z-scores comparable to those of template structures (3RGW, 3UQY) could not be obtained. However, z-score levels were below −1, indicating a significant confidence level.

The results of homology modeling of all the studied hydrogenases confirm the presence of 4Fe-3S clusters in the position proximal to the active site. Generally, homology modeling positioned FeS-atoms of the built models close to corresponding cysteine residues, and the cysteine residues of models and templates could be superposed quite well. However, some artifacts occurred, such as rotation of cysteine SH groups from the Fe atoms. Figure 1 shows an example of such a problem that appeared during modeling the small subunits.

This problem can be solved either by constraining the dihedral angle around Cα-Cβ atoms of cysteine residue or by manual rotation of the cysteine residue after modeling; nonetheless, the combination of primary and tertiary structure alignment gives quite a definite evidence on the same structure of proximal FeS cluster as in *E. coli* hydrogenase-1 or HoxKG-hydrogenases of *R. eutropha* and *Hv. marinus*, because six cysteines in corresponding positions and oriented similarly in space, instead of four cysteines in “classic” hydrogenases (like NiFe-hydrogenases of sulfate-reducing bacteria or *Allochromatium vinosum*) must reduce the number of inorganic sulfurs required to coordinate iron by 1, since the valence of inorganic sulfur (S^2−^) is 2, while the valence of the cysteinyl group (–CH_2_–S^−^) is 1.

The other FeS clusters, medial and distal, do not differ significantly from the clusters of “classic” hydrogenases.

### 2.2. Microenvironment of Proximal FeS Cluster

When considering the oxygen tolerance provided by the 4Fe3S cluster, one should always keep in mind that the nature of its microenvironment could possibly affect its ability to reduce the oxidized active site.

First of all, one of the most important residues in the small subunit is a glutamic acid residue. All the studied hydrogenases possess a very conserved motif, 73-LAVE-76 (numbering according to HupS from *Thiocapsa roseopersicina*, mature form). The residue E76 corresponds to the E76 residue, which was shown to shift closer to the iron atom during interaction with oxygen [38]. This residue’s sidechain was close to the Fe atoms (around 4.5 Å) in all the studied models.

The multiple alignment of the large subunits of the studied hydrogenases with known oxygen-tolerant enzymes showed a lower amino acid conservation (Figure 2: yellow color shows difference from HoxG/HyaB/HydB; cyan color shows identity).

It is interesting to note several things. First of all, only four of the six studied enzymes possess both residues highlighted by Bowman et al. as determinants of oxygen stability (measured experimentally as the reversibility of current after injection of oxygen into the hydrogen-feeding hydrogenase electrode), glutamate E73, and histidine H229 [16].

Moreover, only one of the studied hydrogenases has a valine residue which is conserved in the crystallized oxygen-tolerant hydrogenases, whereas other purple bacteria have the more hydrophilic threonine residue instead. The role of this residue should also be investigated, because this residue is present in the corresponding position in oxygen-sensitive hydrogenases of sulfate-reducing bacteria.

According to Bowman et al., substitution of the glutamate residue by alanine led to a significant drop of stability vs. oxygen; however, little is known about substitutions of this residue to glutamine, although it seems to be able to affect the electrochemical behavior of the FeS cluster.

So, two polar variants can be selected among the modeled hydrogenases: hydrogenase from *Rvi. gelatinosus* is the one resembling well-known oxygen-tolerant membrane-bound hydrogenases, like HoxKG from *R. eutropha (Alcaligenes eutrophus* and *Cupriavidus necator)*, whereas hydrogenase from *Rps. palustris* has the maximal degree of difference from such hydrogenases (Figure 3).

The three residues differing in those two hydrogenases (or residue pairs) can be arranged in a series according to the decrease of their possible effect on the electrochemical properties of the clusters and thus the oxygen tolerance. The first one should be E/Q: Its sidechain is only 10 Å from the closest iron atom of 4Fe3S cluster, as well as from the active site, so the presence or absence of the carboxylic group close to the proximal cluster looks like a factor affecting oxygen tolerance.

The next residue pair in the series is V/T: it is farther from the active site and 4Fe3S cluster, 11.5–12.5 Å when measured from the differing (methyl or hydroxyl) heavy atom, and its action could hardly be explained so explicitly.

The last pair is F/Y, where the closest carbon from the aromatic group is 20 Å from the mentioned redox ligands and 16–17 Å from the medial 3Fe4S cluster.

Besides the physical or at least geometrical basis, there should be statistical confirmation of the statement about the possibility of mediation of oxygen tolerance by certain residues.

Analysis of large subunits from HydDB database (group 1d) [39] showed that when aligning 215 amino acid sequences from the 1d group, 214 of them contained the 68-WAFVERICGVC-78 motif or its homolog (numbering as in the PDB file of HoxKG from *R. eutropha*, 3RGW); one sequence was possibly just a fragment. The consensus is shown in Figure 4, and the exact residue number for each alignment position is listed in Appendix A.

As for the second conserved motif (223-FGGKNPHPNYLVGG-236), the consensus sequence is presented in Figure 5, and the exact residue number for each alignment position is listed in Appendix A.

Although the residues specific for the crystallized oxygen-tolerant hydrogenases (V71, E72, and F223) really prevail in the group, the first two of them are not present in such overwhelming majority to be called “absolutely required” for oxygen tolerance. However, the results of Bowman et al. showed that the E73A substitution in *S. enterica* hydrogenase-5 led to the appearance of irreversible components in the inhibition process of hydrogenase electrode by oxygen, whereas the native enzyme displayed almost 100% restoration of activity. On the other hand, the aforementioned hydrogenase from *A. aeolicus* also possesses 4Fe3S cluster (both supernumerary cysteines are present in the small subunit), but it has a somewhat different sequence in the large subunit, WAFTQRICGVC [16].

### 2.3. C-Termini of the Small Subunits

The C-termini of the small subunits comprise alpha-helical motifs. Their hydropathicities vary, but for all the studied hydrogenases, there could be observed an increase of hydrophilicity from the N-terminus to the C-terminus of the C-end fragment.

To test the ability of the C-end fragments to form transmembrane anchors, they were analyzed on the TMHMM Server 2.0 [40]. The analysis showed a clear tendency to form a transmembrane helix in *Rba. capsulatus*, *Rba. sphaeroides*, *Rps. palustris*, and *Rsp. rubrum*, and this helix appears to be from the 15th to the 37th residue of the selected hydrogenase fragments. The multiple alignment of their C-termini is shown below (Figure 6).

The results of the alignment with *E. coli* hydrogenase-1 (PDB ID: 4GD3, and 1G94 [41,42]) showed that the level of identity is too low to model the C-termini via homology modeling; the prediction of transmembrane helices by TMHMM server showed around 70% of overlap between predicted and experimentally observed helices for *E. coli*, which can be reasonable enough to use the data from the TMHMM server as information confirming the transmembrane orientation of the helices of the modeled hydrogenases. In the case of the hydrogenase from *Rvi. gelatinosus*, a region showed the propensity to be a transmembrane alpha-helix rather than extracellular or intracellular part of the protein; but the value of the score was low (below 0.6), whereas for *Rhodobacter*, *Rhodopseudomonas*, *Rhodospirillum*, and *E. coli*, the values were above 0.8. It is interesting to note that the substitution of four arginine residues in *Tca. roseopersicina* by the corresponding uncharged residues from *Rhodobacter* (Q, L, V, and A) makes the server predict a transmembrane helix in the derived variant as well. As for *Rvi. gelatinosus*, substitution of just a single lysine residue to alanine or isoleucine led to prediction of transmembrane helix there. The geometry of the C-terminal fragments extracted from full-size models of the enzymes is shown in Figure 7.

The possibility of formation of transmembrane helices is debatable, since three out of four of the helices predicted by TMHMM were kinked during modeling (20 ns simulations in water before using as templates) (see Figure 7c,d,f), which is evidence of the labile links in the moieties of the C-terminal fragments. Probably, the presence of glycine residues makes these helices labile. Another possibility could be a prediction error of the TMHMM server itself, which was confirmed for the case of *E. coli* hydrogenase: the server could predict the exact positions of the transmembrane regions of the helices incorrectly, as in case of *E. coli* hydrogenase (Figure 6), and their actual positions could be closer to the C-termini. The length of the predicted transmembrane helices (20 Aa) is enough to span the entire membrane.

It must be realized that the exact position of the transmembrane region varies even between the subunits of *E. coli* hydrogenase, so one can imagine the possibility of sliding the transmembrane helix across the membrane bilayer. Hence, the variations in the predictions made by the TMHMM server could be physiologically irrelevant.

### 2.4. Full-Size Models of the Hydrogenases

The presence of long helical moieties in the full-sized small subunits increased their normalized DOPE z-scores significantly, leading to the results that cannot be considered as good as for globular proteins. Since the z-DOPE levels for many enzyme models were above −1 for the small subunits, z-DOPE levels for the aligned parts (i.e., parts having homologous experimental structures) in full-size models were also taken into account.

Nevertheless, one should keep in mind that DOPE statistical potential was developed and calibrated as a scoring function on a sample of cytosolic globular proteins, so the high z-DOPE levels are in agreement with the suggestion of possible role of the C-end fragments as membrane anchors of hydrogenases.

The normalized DOPE z-scores of the aligned parts of the hydrogenase subunits after energy minimization displayed values corresponding to high confidence levels. The results of DOPE assessment for hydrogenase subunits are summarized in Appendix A. Analysis of the X-ray structure of the full-size hydrogenase from *E. coli* (PDB ID: 4GD3) showed that the structure of its small subunit has a quite poor z-DOPE score (−0.476 to 0.513 before energy minimization, −0.547 to 0.619 after), so the results for the models and for the X-ray structure do not differ by much.

Overall views of the full-size models of the studied enzymes are shown in Figure 8 (see larger views in Appendix A).

### 2.5. Intrasubunit and Intersubunit Interactions

The results of intersubunit interaction assessment showed that the main contributors to the stability of the intersubunit interface are hydrophobic contacts; however, there was no statistical significance of the intersubunit interactions in the studied enzymes.

Calculations of hydrogen bonds led to very high dispersions and showed no reliable differences between the enzymes. Other interactions, such as π–π (aromatic–aromatic), π–cationic, and aromatic–sulfur, did not show reliable differences and made only slight contributions to the overall interaction network of the enzymes. There were three types of interactions that showed statistically significant differences between the different enzymes: hydrophobic contacts in the large subunits, ionic pairs in the large subunits, and ionic pairs between subunits (Figure 9, Figure 10 and Figure 11). The numerical data are listed in Appendix A.

It should be noted that models of *Rhodobacter* hydrogenases have significantly lower number of hydrophobic interactions between the residues of large subunits (Figure 9). This could have an effect on the stability of the enzymes, either at elevated temperatures or in presence of solvents other than water. Since the large subunit is linked by fewer metal–protein interactions than the small one, its stability could be the limiting factor during denaturation of hydrogenases.

The results of ionic interaction calculations allow for the division of the modeled enzymes into two groups, on the basis of number of ionic interactions in their large subunits, “high-ionic” (*Tca. roseopersicina*, *Rba. capsulatus*, *Rba. sphaeroides*, and *Rvi. gelatinosus*) and “low-ionic” (*Rsp. Rubrum* and *Rps. palustris*). This statistically significant difference could also affect some properties of the enzymes, for example, thermal stability at different salt concentrations, which could be important when applying these hydrogenases in fuel cells with a high electrolyte ionic strength. It must be considered that the number of ionic interactions is usually significantly increased during energy minimization in the YASARA force field, as follows from the results of the present calculations.

As for intersubunit interactions, the low number of ionic pairs in HupSL from *Tca. roseopersicina* reported by Szilagyi and colleagues [26] was also observed while using models before energy minimization. However, after this procedure, the distributions of intersubunit ionic pair numbers became wider, so this result apparently lost its statistical significance.

### 2.6. Molecular Tunnels: Possible Oxygen Pathways

The effect of differences in the intramolecular tunnel structure on the oxygen stability of the studied hydrogenases cannot be considered as definitely significant, because the diffusion kinetics is not clear; however, it could be argued that some differences favor some of the studied hydrogenases.

The tunnel fine structure depended on energy minimization procedure (was carried out or not), because many tunnels vanished or became narrower after energy minimization. When starting points were selected between the NiFe active site and the 4Fe3S cluster, tunnel structure was very unstable; more stable mapping results were obtained during automatic selection of tunnel structure. There were several tunnels with a high hydropathicity level which could act as gas-accession channels. When setting the “bottleneck tolerance” parameter to 0 (i.e., all the tunnels with the narrowest point narrower than 1.2 Å were removed), there appeared differences between the studied enzymes (models and X-ray structures from the PDB). For instance, only one tunnel close to the active site was mapped in the case of HupSL from *Tca. roseopersicina* and HyaAB from *E. coli* (PDB ID: 3UQY) when the bottleneck tolerance value was set to zero; two tunnels were mapped in *R* hydrogenases, but their level of similarity was close to the upper limit (0.7), so they could be treated as one tunnel. The hydrogenase from *H. marinus* (PDB ID: 3AYX) showed no tunnels close to the active site in such conditions.

Some of the tunnels start really close to the active site; an example of such a tunnel network is shown below (Figure 12). Results of tunnel mapping after automatic starting point selection are given in Appendix A (ijms-21-00366_tunnels_zerotolerance.zip).

Since tunnel mapping is quite a complicated process, which is dependent on multiple factors, it required verification by comparing with experimental data of Kalms and colleagues [38,43]. For this purpose, krypton-assessed (PDB ID: 5D51) and oxygen-derived (PDB ID: 5MDL) results of tunnel mapping in *R. eutropha* membrane-bound hydrogenase were taken. As there are gas molecules present in these PDB files, tunnel-lining residues were calculated by setting a 4 Å distance cutoff from the gas molecules. The detailed results of comparison of hydrogenases from *Rsp. rubrum* and *R. eutropha* are listed in Appendix A; the level of tunnel similarity allows us to make the preliminary conclusion that the tunnel structure is quite similar between two enzymes selected for comparison; the level of tunnel similarity in the PDB files of *R. eutropha* hydrogenase shows some level correlation of mapping with experimental data. Also, lining residues in the modeled enzyme showed some similarity with the experimental structure. However, it must be noted that the diversity between the studied hydrogenases does not allow their subdivision into groups correlating their oxygen tolerance with tunnel structure; other factors, such as the ability to exchange electrons between the small subunits in the oligomeric state, appear to be more essential in determining oxygen tolerance. Prediction of this ability is described in the next section.

### 2.7. Possible Oligomerization: Protein–Protein Docking Results

According to data obtained by combined ClusPro docking and PISA assessment of binding energies, mutual binding affinity in the calculated complexes is higher than in the reference structure (*E. coli* hydrogenase, PDB ID: 4GD3). However, it should be noted that the native *E. coli* hydrogenase-1 is bound to the membrane, and it also interacts with a cytochrome molecule; thus, the strong direct interaction between two hydrogenase molecules is not required. As for distances between FeS clusters of adjacent hydrogenase small subunits in (HupSL)_2_-heterotetramer, for most cases, they were comparable to those observed in the oxygen-tolerant hydrogenase oligomers, but for the enzymes from *Rba. sphaeroides*, the distances were close to 30 Å. This does not imply an impossibility of electron transfer, but the role of surrounding aromatic residues as possible “hopping sites” for electrons (intermediate states which could exchange electrons or holes with the FeS clusters) should increase. Figure 13 shows the overview of docking complexes and close view of FeS clusters of two adjacent small subunits in case of two HupSL-HupSL dimers, from *Rba. capsulatus* and *Rba. sphaeroides* (see details for all the studied enzymes in Appendix A).

The results of docking-based oligomerization modeling show that the studied enzymes should have the ability to exchange electrons between the FeS clusters of adjacent small subunits, so oxygen tolerance should take place.

### 2.8. Interaction with Electrode Compounds: Docking of Small Molecules

Structural models of docked ligands are listed in Figure 14.

The results of molecular docking of conductive compounds showed that the modeled hydrogenases are comparable to the oxygen-tolerant hydrogenases with known X-ray structure. The specificity and affinity of binding in blind docking analysis were similar to the hydrogenases of hydrogen-oxidizing bacteria (detailed data in Appendix A).

A thorough analysis of the obtained docking complexes revealed the different modes of binding, which could be divided into “productive” and “non-productive” complexes. “Productive” complex means that the distance between FeS clusters and the ligand is less than 20 Å (this is quite an arbitrary value, but in some way it corresponds to the possibility of electron transfer); the ratio of occupancy between these complexes (a fraction of productive complexes) could roughly reflect the probability of binding into productive and non-productive complexes in the real solution (suspension) or on the electrode surface. Moreover, the distance from the closest ligand atom to the FeS cluster could serve as a qualitative indication of electron transfer rate. An example of graphene oxide binding to HupSL from *Rba. sphaeroides* is shown in Figure 15.

Among all the enzymes assessed by docking, models of HupSL from *Rba. sphaeroides* and *Rvi. gelatinosus* before energy minimization exhibited the highest fraction of productive complexes with graphene oxide (15/20 and 13/20, respectively; see Appendix A); after energy minimization in solution, the highest productivity (11/20) was observed for the interaction of *Rsp. rubrum* and *H. marinus* hydrogenases with neutral red trimer (Appendix A).

The results of blind docking have certain limitations. The affinity values cannot be treated as exact ones, since the sidechains of the binding sites should be treated as flexible ones when simulating the exact binding. The difference between productivity of ligand interactions with enzyme models before and after energy minimization on YASARA server could reflect the difference in applicability in various media. The enzyme models before YASARA energy minimization were optimized by in vacuo molecular dynamics at the final stages of modeling process; hence, they could better describe the structure and behavior of the proteins in dry, non-aqueous (gaseous) environments. As gas-breathing fuel cells seem to be quite promising from the point of view of current densities, the results obtained by models in vacuo should be taken into account. The models after energy minimization were processed in aqueous solution, so they would better describe the enzymes in more conventional media like solutions or solution-embedded electrode surfaces.

The chosen criterion for productivity of the complexes (20 Å distance cutoff) might seem not to be strict enough when compared to native distances between electron-transferring subunits in hydrogenases or hydrogenase complexes. For example, the NiFe–FeS distance is 13.7 Å, and the same distance is observed between the distal FeS cluster and the heme aromatic group in the hydrogenase-1: cytochrome complex of *E. coli* (PDB ID: 4GD3); distances between distal FeS clusters in the *S. enterica* hydrogenase oligomer (PDB ID: 4C3O) are also 14–16 Å. On the other hand, it was shown that even a distance of 32.3 Å could be enough for direct electron transfer [44]. However, this resulted in the microampere current densities, whereas design of high-current density electrodes will require shorter distances from Fe to the surface.

## 3. Discussion

The HupSL-hydrogenases of purple phototrophic bacteria have been neglected for quite a long time. Since their discovery in 1950s–1980s [45,46,47], they were considered mostly viewed as something to get rid of in order to improve hydrogen-producing strains of purple bacteria (for example, [48]). However, some works did really employ the activities of these enzymes. For example, there were two works on hydrogen production by *E. coli* cultures transformed with Hup-hydrogenase genes from *Rba. sphaeroides* [49] and *Rps. palustris* [50]. Interestingly, both these cases provided significant improvement of hydrogen production by *E. coli*; so, although the catalytic bias of the membrane-bound hydrogenases toward hydrogen oxidation is well-known and their main physiological role is the utilization of hydrogen evolved in the nitrogenase reaction [47,48], the “uptake” nature of these hydrogenases is not physiologically irreversible.

The aforementioned inhibition study of inhibition of *Rba. capsulatus* HupSL by diphenylene iodonium [28] showed that there should be two binding sites for different electron acceptors; diphenylene iodonium was also demonstrated to inhibit hydrogen production by *Rhodobacter sphaeroides*. The authors suggested that such inhibition was due to the interaction of this compound with hydrogenase, rather than with nitrogenase, thus providing extra evidence for HupSL-dependent hydrogen production [51]. Taken together, these data show that the electron flow in HupSL hydrogenases is quite a flexible matter to study, and structural models obtained in the current work would help to understand this complex subject more clearly.

A few words must be said about *Rsp. rubrum* hydrogenase. It has been shown to be present preferably in culture liquid (overall activity in culture liquid being 10 times higher than in disrupted cell extracts), and reasonable production amounts of this enzyme were achieved by simple addition of EDTA to the culture medium. This could be explained by its susceptibility to cleavage by a metalloprotease. Adams and Hall demonstrated that this enzyme is deactivated by air (half-life of 7 days in air vs. 12 days under N_2_ atmosphere) [31]. However, Kakuno et al. showed that this enzyme can be stabilized by EDTA and high salt concentration (no detectable loss of activity in 6 months at salt (NaCl, KCl, and CsCl) concentrations above 0.7 M under air; the enzyme tolerated up to 4 M of NaCl) [32]. This stabilization cannot be explained just by a decrease in oxygen solubility, since it decreases only by 1.5 times. Although the data obtained by Kakuno and colleagues do not evidence that it was exactly Hup-hydrogenase, later data by Manness clarified that, among the three hydrogenase activities, two are inducible, and Hup hydrogenase has the highest hydrogen uptake activity [52]. The low number of ionic pairs shown in the present study for the catalytic subunit of this enzyme could explain its stability toward high salt concentration. Thus, this makes the *Rsp. rubrum* enzyme a good candidate for experimental assessment in electrochemistry in solutions with high conductivity.

*Rvi. gelatinosus* was studied from the point of view of its ability to catalyze water–gas shift reaction [48]. Another study [29] showed that HupSL deletion is an effective way of providing hydrogen production from CO, but nothing could be said about the properties of HupSL hydrogenase.

It should be noted that previous studies did also point to supernumerary cysteine residues in the small subunits of *Rvi. gelatinosus* hydrogenase [24], but this study contained just a remark without detailed examination of the structure of this enzyme. Now, it is time to look at this enzyme as one of the most promising candidates for the role of novel oxygen-tolerant catalyst (among purple bacteria), since it has the highest similarity of amino acid residues with crystallized oxygen-tolerant enzymes from chemotrophic bacteria. The presence of valine instead of threonine could lead to increased hydropathicity in the region of the 4Fe3S cluster. Its good binding to graphene oxide (although shown for model before energy minimization) could be advantageous in electrode fabrication, and the same can be argued for HupSL from *Rba. sphaeroides*.

As for *Tca. roseopersicina*, it is the only sulfur bacteria among all the organisms covered in the present modeling study. Its Hup-hydrogenase was not studied as intensely as its Hyd- (or Hyn-) hydrogenase. Some studies could lead to proposals that this enzyme is not promising from the biotechnological point of view. For example, it was shown to lose its activity irreversibly under air in just a day [27]. Since then, it has become mostly the subject of basic research exploring regulation of its synthesis [53,54], but, from there, some interesting information could arise. Several works demonstrated the susceptibility of Hup-hydrogenases toward cold denaturation ([43,53,55] p. 83). However, the techniques used in these works did not exclude possible proteolysis (the spectra of used protease inhibitors were insufficient to suppress it completely; moreover, EDTA helped to stabilize the hydrogenase from *Rhodospirillum rubrum*, as was mentioned earlier), so these results should be regarded quite critically. For example, the proteomes of several purple non-sulfur bacteria contain homologs of proteases which are more active in a cold environment. These proteases are DegP-like protein of *R. sphaeroides* 2.4.1 (Uniprot ID: Q3IX80) and *R. capsulatus* SB1003 (Uniprot ID: D5ALS1), both sharing 37% of identical residues with DegP protein from *E. coli* (Uniprot ID: P0C0V0) [56]. Their activities should be investigated thoroughly in cases of preparation of proteins from these bacteria using cold temperatures or proteolysis inhibition; perhaps specific inhibitors of these enzymes should be designed in order to prevent the proteolysis of hydrogenases.

Rather poor z-DOPE levels of the full-size models of Hup hydrogenases from *Rba. capsulatus*, *Rba. sphaeroides*, *Rps. palustris*, and *Rsp. rubrum* can be explained by their hydropathicity and tendency to form membrane anchors, since DOPE statistical potential was calibrated on globular hydrophilic proteins. In other words, the quality of the models assessed by z-DOPE is somewhat correlated with the results of the TMHMM predictions for the presence of transmembrane helices at the C-ends of the studied enzymes. Moreover, the z-DOPE levels are comparable to those obtained for hydrogenase-1 from *E. coli* (PDB ID: 4GD3, 6G94).

The tunnel structure in hydrogenases is quite a complex subject to study, since one should always keep in mind its flexibility. Structure of tunnels in solution could be most accurately reflected by the results on energy-minimized structures; however, when taking into account gaseous electrodes for air-breathing fuel cells, one can suppose that the protein structure, and thus its tunnel structure, could be more accurately approximated by the crystal structure or the frames of in vacuo molecular dynamics, since the enzyme is placed in an almost dry atmosphere or on the hydrophobic surface of carbonaceous material [57]. Nevertheless, the structure of the tunnels mapped in the current work was very diverse, and more reliable results addressing the issue of possible oxygen stability were achieved by protein–protein docking, showing that the FeS clusters in oligomers of the modeled enzymes can be close enough to provide intersubunit electron exchange between the small subunits.

When using molecular docking as a method for assessing the prospects of a certain redox enzyme in applied electrochemistry, one must take into account the need for formation of a complex during the contact of the enzyme with the electrode surface. For instance, the plane of graphene oxide must be oriented in such way that it could be expanded to model a plane of carbon; in other cases, when the plane is embedded into a protein pocket, it could model the behavior of the enzyme just on the edge of the electrode surface.

The increase in the affinity of hydrogenases to neutral red oligomer with oligomerization degree shows that neutral red dye could also be a promising immobilization agent for these hydrogenases. Since it showed its application prospects with HydSL hydrogenase from *Tca. roseopersicina* [55], there is definitely a reason to test it in a series of other hydrogenases. In this work, the hydrogenase from *Rsp. rubrum*, as well as the experimentally determined *H. marinus* enzyme, showed the highest degree of specificity for binding neutral red trimer.

Besides purely geometrical criteria of hydrogenase sorption efficiency, such as the distance from the FeS clusters to the electrode and the angle between the graphene oxide plane and the enzyme surface, there is another parameter that was not covered in the current work, namely dipole moment. It was shown to be an important factor of efficient interaction of membrane-bound hydrogenases from *R. eutropha* [44] and *A. aeolicus* [58] with electrode surfaces; however, high-current density fuel cells seem to require the shortest distance from the FeS cluster to electrode surface, with dipole moment being the second factor. Moreover, short-range interaction between the electrode surface and the FeS cluster should increase the impact of local electrostatics of cluster microenvironment, thus decreasing the role of dipole moment of the HupSL dimer as a whole. Nevertheless, dipole moment is definitely an interesting subject for further study.

The biotechnological potential of Hup-hydrogenases should be analyzed not only in vivo, but also in vitro. Since 4Fe3S-cluster-containing hydrogenases are treated as “oxygen-tolerant”, such tolerance should be examined in Hup-hydrogenases from purple bacteria. Despite the impressive success achieved so far with studies of *E. coli* and *R. eutropha* hydrogenases, there is still no answer to the question of which hydrogenase is “the best” for hydrogen fuel cell application, meaning that the whole diversity of probable oxygen-tolerant hydrogenases must be examined in comparative studies, and an assessment of their strengths and weaknesses should lead to creation of the optimal catalyst for future energy applications.

## 4. Materials and Methods

Sequences for homology modeling of these enzymes were found in NCBI Protein database. According to modern hydrogenase classification proposed by Sondergaard et al. [39], they were assigned to group 1d of NiFe hydrogenases (Aerobic uptake hydrogenases, oxygen-tolerant, possessing [NiFe]-center, 1 × [4Fe3S] cluster, 1 × [3Fe4S] cluster, 1 × [4Fe4S] cluster, interacting with b-type cytochrome). The used sequences are listed in Table A1 (Appendix B).

Before using the sequences in MODELLER, they were preprocessed based on literature data and multiple alignments. The large subunits were truncated to the last C-terminal histidine residue, since the final step of hydrogenase maturation is cleavage of C-terminal peptide [59]; the sequences of large subunit C-termini are listed in Table A2 (Appendix B).

As for the small subunits, the situation was a bit more complicated. Since the small subunit precursors contain twin-arginine motifs, and they were described as being subject to proteolysis accompanying targeting them to periplasm, their N-termini were analyzed to find out the proteolysis sites. The analysis was conducted on three websites, namely SignalP, TatP, LipoP [60,61,62]. All the three sites did robustly predict cleavage between alanine and methionine or alanine and leucine in the N-terminal part of the small subunits. The signal peptides cleaved off the small subunits, and their scores for probabilities of cleavage site presence are listed in Table A3 (Appendix B).

### 4.1. Homology Modeling

Homology modeling was carried out with the MODELLER program package [35,63,64]. Templates for homology modeling were searched by using BLAST online service [65], and the alignments produced were taken as the basis for writing alignments for using in MODELLER.

Several templates were selected for homology modeling (PDB IDs listed): 3AYX (membrane-bound hydrogenase from *Hydrogenovibrio marinus*), 3RGW (membrane-bound hydrogenase from *Ralstonia eutropha* H16), 3UQY (membrane-bound hydrogenase-1 from *Escherichia coli* K12), and 4C3O (*Salmonella enterica* Serovar Typhimurium LT2); after a series of initial modeling runs, two templates, 3RGW and 3UQY, were kept, whereas all other templates were discarded due to poor z-DOPE levels of the derived models.

Since MODELLER allows multisubunit modeling, the sequences present in alignments comprised sequences of both large and small subunits of the hydrogenases of interest. The ligands were included into the models as bulk rigid bodies, i.e., they were not recognized by MODELLER as atoms possessing their own forcefield parameters, except Mg, which was specially designated by “$” sign and treated like a CHARMM27 atom.

Several strategies implemented into MODELLER were combined to increase the confidence level of the homology models: (1) VTFM-optimization; (2) conjugated gradient optimization; and (3) in vacuo molecular dynamics. To find the optimal optimization protocol, the optimization procedure (very thorough VTFM optimization (autosched.slow) and very slow molecular dynamics (md.refine = very_slow) was repeated from 1 to 5 times, with 100 models being produced for each run; the 10 best models were taken for each enzyme out of 500 produced models.

The confidence level of the models was assessed via normalized DOPE z-score (z-DOPE) [35] estimation, and the values below −1 were considered as very good and reliable, according to the recommendation of the MODELLER development team. The key value for sorting the models was the z-DOPE of a small subunit (aligned part of the subunit only).

### 4.2. Ab Initio Modeling

QUARK ab initio modeling server [66] was used in the work for modeling the C-terminal parts of the small subunits of these enzymes. The data obtained from these servers were collected and ranked by the server itself based on their TM-score; they were used as templates for building 100 models in MODELLER with optimization by high-temperature molecular dynamics in vacuo, and the best models assessed by the normalized DOPE z-score were used for the following MD simulation in water.

### 4.3. Molecular Dynamics Simulations

To improve the quality of the ab initio predicted models of the C-terminal parts of the small subunits, molecular dynamics simulations in explicit water during 20 ns were performed in GROMACS [67]. The initial files for molecular dynamics in GROMACS were produced from MODELLER calculations, where the results from QUARK web server were used as templates for the homology modeling calculations, and the final models were selected based on their lowest z-DOPE scores from 100 models of the fragments.

For better agreement between MODELLER and GROMACS simulations, CHARMM22 force field [68] was selected for all the steps of molecular dynamics in GROMACS.

The boundaries for all the simulations were periodic.

Energy minimization was performed by the steepest descent algorithm until the maximum force was less than 1000 kJ/mol/nm. Long-range electrostatic interactions were calculated by the Particle Mesh Ewald (PME) method; short-range electrostatic and van der Waals cut-off radius were 1 nm; SPC water model was used for solvation of the protein in a cubic box extended by 1 nm from the protein; and 100ps NVT (in modified Berendsen thermostat [69]) and NPT (in modified Parrinello–Raman barostat [70]) calculations were followed by 20ns production molecular dynamics.

The resulting frames were saved every 4 ns, so that 5 conformers of the C-terminal part of the small subunits were used in modeling the full HupSL dimers.

### 4.4. Assessment of the Role of Ab Initio Modeled Parts in Membrane Anchoring

To predict the tendency of the studied fragments to form transmembrane alpha-helices, they were analyzed on the TMHMM server, version 2.0 [40] (www.cbs.dtu.dk/services/TMHMM/).

### 4.5. Full-Size Hydrogenase Modeling

Full-size hydrogenase modeling was carried out in MODELLER, using the alignments expanded by addition of the C-terminal fragments to modeled hydrogenase sequences; the conformers of C-terminal domains obtained from molecular dynamics in explicit water were assessed by normalized DOPE z-score, and those having the lowest values were used as templates for the C-terminal parts. The optimization scheme was the same as before, i.e., VTFM-optimization and high-temperature molecular dynamics in vacuo.

### 4.6. Intrasubunit and Intersubunit Interaction Assessment

Intrasubunit and intersubunit interactions were estimated on the Protein Interaction Calculator Web server (http://pic.mbu.iisc.ernet.in/) [71].

Electrostatic interactions were assessed by the same criterion as reported by Szilagyi and co-workers: the cutoff for centroid–centroid distance was set to 6 Å [26].

Hydrophobic contacts were counted as the number of atom–atom distances less than 5 Å between aliphatic or aromatic carbons of the following residues: Ala, Val, Leu, Ile, Met, Phe, Trp, Pro, and Tyr.

Aromatic-aromatic interactions were calculated as the number of distances between aromatic group centroids in the range from 4.5 to 7 Å.

### 4.7. Intramolecular Tunnels

To find out possible pathways for gas diffusion inside the studied hydrogenases, intramolecular tunnels were mapped on MOLE2.5 server (mole.upol.cz) [72]. First, NiFe active site was used as starting point for the tunnels; automatic selection of starting point was chosen in further calculations. Hydrogen atoms were ignored, whereas HETATM groups were taken into account. All the other settings were taken by default: origin radius 5 Å, surface cover radius 10 Å, bottleneck radius 1.2 Å, bottleneck tolerance 3 (changed to 0 in the second round of mapping), maximal tunnel similarity 0.7. Tunnel mapping was performed in both non-minimized and minimized models of Hup-hydrogenases and compared with the X-ray structures of hydrogenases from *Hydrogenovibrio marinus* (3AYX), *Ralstonia eutropha* (3RGW), *Escherichia coli* (3UQY), and *Salmonella enterica* (4C3O).

### 4.8. Protein–Protein Docking and Oligomerization Assessment

The obtained models (HupSL-dimers) were submitted to ClusPro web server [73]. After obtaining the results, they were analyzed on the PISA web server [74]; the oligomeric structure of *E. coli* hydrogenase (PDB ID: 4GD3) was taken as reference enzyme. The best ClusPro variants were superposed with the initially submitted models to extract heteroatomic groups in correct positions. Distances between FeS clusters were measured as the shortest distances between Fe atoms. Aromatic residues between the Fe atoms of adjacent small subunits were regarded as “hopping points” for electron.

### 4.9. Docking of Small Molecules

The hydrogenase models were prepared for docking in Autodock Tools [75,76]. Polar hydrogens were added and AD4 atom types were assigned during receptor preparation. The following ligands were used in the docking study: neutral red monomer, dimer and trimer, and graphene oxide as a model of carbon electrode surface. Oligomers of neutral red were modeled according to Paulikaite et al. [77]; the graphene oxide molecule was taken from PubChem (PubChem CID: 124202900) and converted into 3D PDB file in Corina (https://www.mn-am.com/online_demos/corina_demo) [78]. Docking was performed in Autodock Vina [79]. Box size for docking was 120 × 120 × 120 Å; the number of complexes was set to 20, and the exhaustiveness parameter was set to 100 (for the case of graphene oxide. The settings were similar to the approach used by Sumaryada et al. [33]).

## 5. Conclusions

To summarize the results covered by the present article, the author should highlight several important conclusions drawn from the work. First of all, the presence of the 4Fe-3S cluster was confirmed by homology modeling of spatial structure in six hydrogenases from purple bacteria. Moreover, these hydrogenases were predicted to be able to oligomerize in complexes providing electron exchange between adjacent small subunits. Next, the heterogeneity of the cluster’s microenvironment was shown; the hydrogenase from *Rvi. gelatinosus* was expected to be much like the common membrane-bound oxygen-tolerant hydrogenases. Third, the C-ends of the small subunits were assessed as possible membrane anchors. Fourth, analysis of a large amount of data gave statistically significant differences between the number of hydrophobic and ionic interactions in the modeled enzymes. Fifth, protein–protein docking predicted the possibility of electron exchange in oligomeric complexes of these enzymes, supporting a hypothesis of their oxygen tolerance. Sixth, molecular docking studies showed basically the same results for the modeled enzymes as for the X-ray structures, with a slight advantage for the HupSL enzymes from *Rba. sphaeroides* and *Rvi. gelatinosus*, in the case of graphene oxide and that from *Rsp. rubrum* in the case of neutral red trimer.

## Figures and Tables

**Figure 1 ijms-21-00366-f001:**
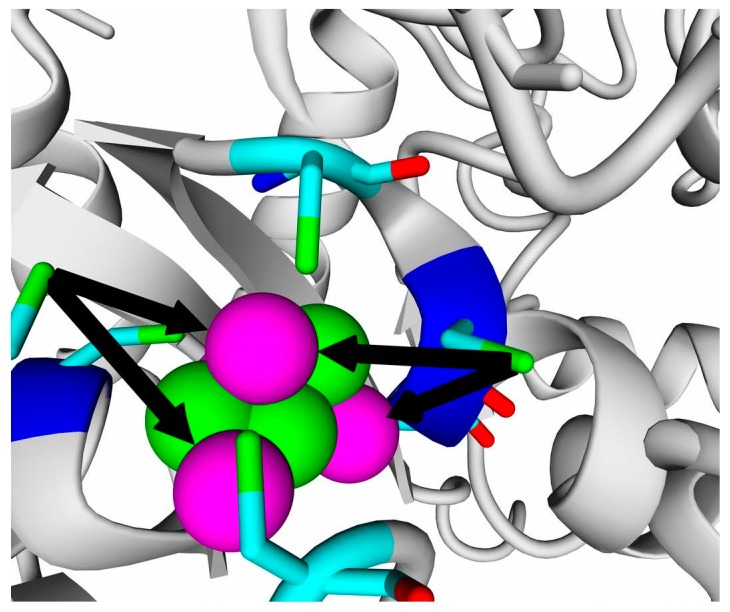
Proximal FeS cluster of *Rba. capsulatus* (model, this work). Two cysteine residues too far from the corresponding Fe atoms are marked by black arrows. Iron atoms are shown as magenta spheres, and sulfur atoms are shown as green spheres. Cysteine residues are colored by element coloring scheme: green, sulfur; cyan, carbon; blue, nitrogen; and red, oxygen.

**Figure 2 ijms-21-00366-f002:**
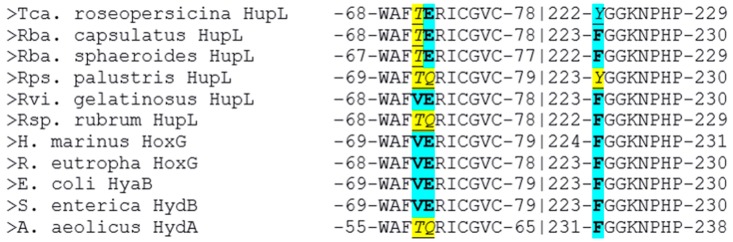
Microenvironment of the FeS clusters of the studied hydrogenases shown in multiple sequence alignment of large subunits. The amino acids found mainly in oxygen-tolerant hydrogenases are colored cyan and shown in bold type; those found in oxygen-sensitive hydrogenases are colored yellow and shown in underscored italic. NB: despite the presence of TQ-motif, hydrogenase from Aquifex aeolicus is also oxygen-tolerant.

**Figure 3 ijms-21-00366-f003:**
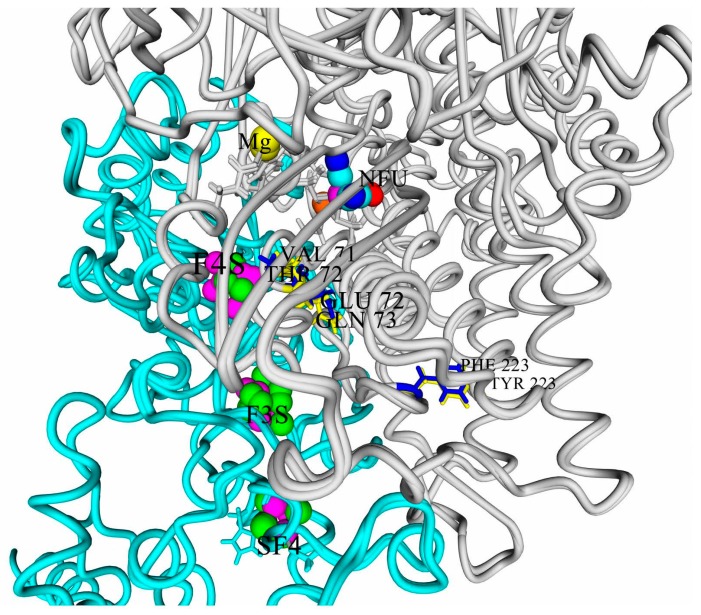
Microenvironment of the clusters of the studied hydrogenases: superposition of hydrogenases from *Rps. palustris* and from *Rvi. gelatinosus*. Most of the atoms of the active site (NFU) and FeS clusters are colored by YASARA element coloring scheme: iron, magenta; sulfur, green; carbon, cyan; oxygen, red; and nitrogen, blue. The nickel atom is colored orange, and the magnesium atom (Mg) is yellow. The large subunits are colored gray, the small subunits are cyan, the residues specific for “oxygen-tolerant” hydrogenases are colored blue (in HupSL from *Rvi. gelatinosus*), and the ones specific for “oxygen-sensitive” enzymes are shown in yellow (in HupSL from *Rps. Palustris*). F4S-proximal 4Fe3S cluster; F3S-medial 3Fe4S cluster; SF4-distal 4Fe4S cluster.

**Figure 4 ijms-21-00366-f004:**
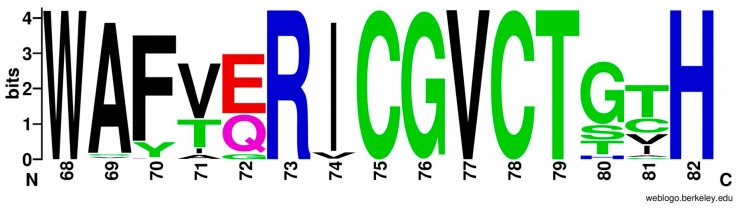
Sequence logo of large subunit consensus for hydrogenases from HydDB database (214 of 215 sequences, residues from 68 to 82). Hydrophobic residues are shown in black, amides are shown in magenta, other hydrophilic neutral residues are shown in green, negatively charged residues are shown in red, and positively charged residues are shown in blue. Ordinate shows information content of the alignment positions in bits (log_2_20 = 4.321928 if the residue is absolutely conserved).

**Figure 5 ijms-21-00366-f005:**
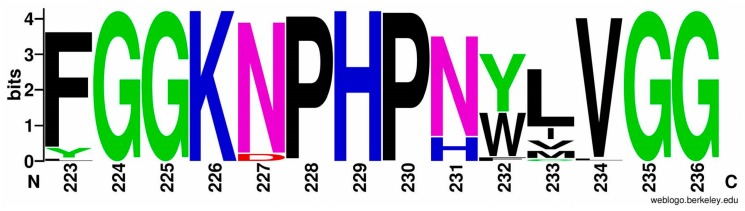
Sequence logo of large subunit consensus for hydrogenases from HydDB database (215 sequences, residues from 223 to 236). Hydrophobic residues are shown in black, amides are shown in magenta, other hydrophilic neutral residues are shown in green, negatively charged residues are shown in red, and positively charged residues are shown in blue. Ordinate shows information content of the alignment positions in bits (log_2_20 = 4.321928 if the residue is absolutely conserved).

**Figure 6 ijms-21-00366-f006:**
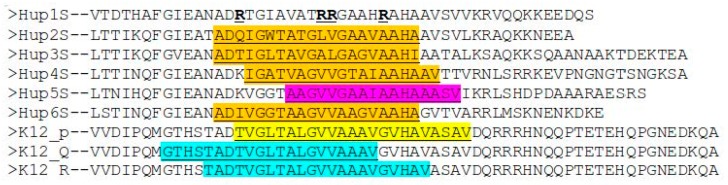
Predicted transmembrane helices in the modeled hydrogenases are highlighted in orange; a region in *Rvi. gelatinosus* showing a slight transmembrane propensity is colored magenta (explanation below); predicted helix in *E. coli* is highlighted yellow, and experimentally observed α-helices are colored cyan. Four arginine residues are shown in bold type: their substitution to corresponding residues from Hup2 leads to prediction of alpha-helix. **Hup1S**-HupS from *Tca. roseopersicina*; **Hup2**-HupS from *Rba. capsulatus*; **Hup3S**-HupS from *Rba. sphaeroides*; **Hup4S**-HupS from *Rps. palustris*; **Hup5S**-HupS from *Rvi. gelatinosus*; **Hup6S**-HupS from *Rsp. rubrum*; K12_p-*Hyd-1 from E. coli*, *TMHMM prediction*; *K12_Q-Hyd-1 from E. coli*, *PDB ID: 4GD3*, *chain Q*, *transmembrane region*; *K12_R-Hyd-1 from E. coli*, *PDB ID: 4GD3*, *chain Q*, *transmembrane region*.

**Figure 7 ijms-21-00366-f007:**
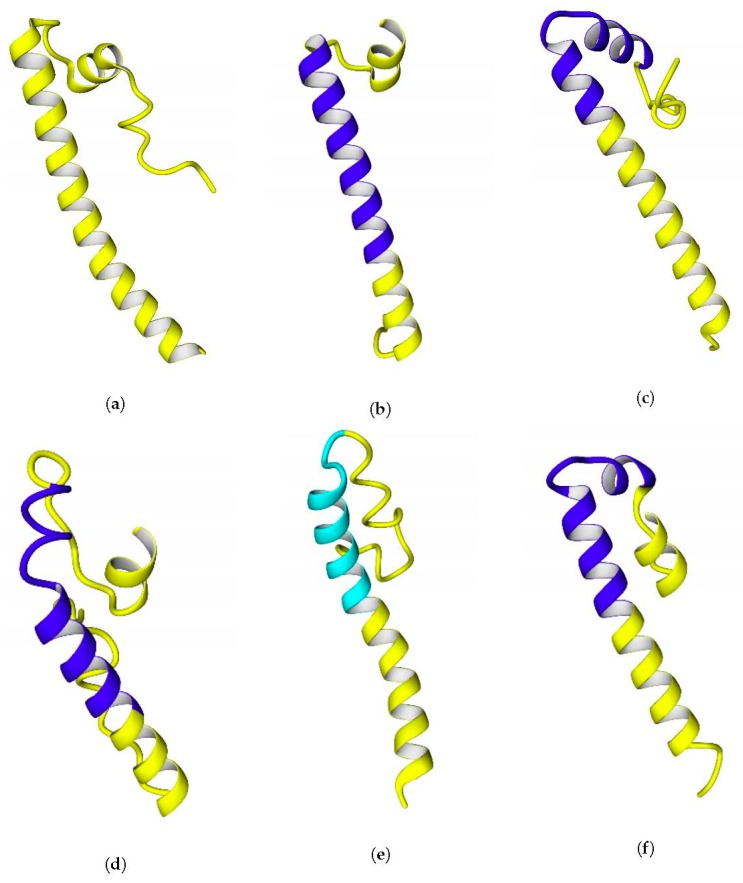
C-terminal parts of the small subunits of the hydrogenases (extracted from final full-size models, see below). Regions corresponding to predicted transmembrane helices are colored blue. The corresponding region of HupSL from *Rvi. gelatinosus* is colored cyan. The C-termini of the helices are positioned at the bottom of each figure. The panels of the figure correspond to the following enzymes: (**a**) HupS from *Tca. roseopersicina*; (**b**) HupS from *Rba. capsulatus*; (**c**) HupS from *Rba. sphaeroides*; (**d**) HupS from *Rps. palustris*; (**e**) HupS from *Rvi. gelatinosus*; and (**f**) HupS from *Rsp. rubrum*.

**Figure 8 ijms-21-00366-f008:**
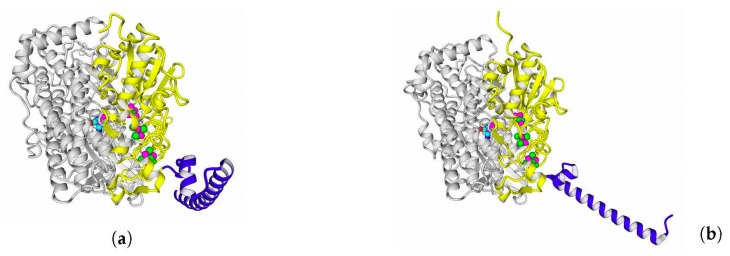
Full-size models of HupSL hydrogenases. Ab initio modeled fragments are colored blue. The panels of the figure correspond to the following HupSL enzymes: (**a**) *Tca. roseopersicina*; (**b**) *Rba. capsulatus*; (**c**) *Rba. sphaeroides*; (**d**) *Rps. palustris*; (**e**) *Rvi. gelatinosus*; and (**f**) *Rsp. rubrum*.

**Figure 9 ijms-21-00366-f009:**
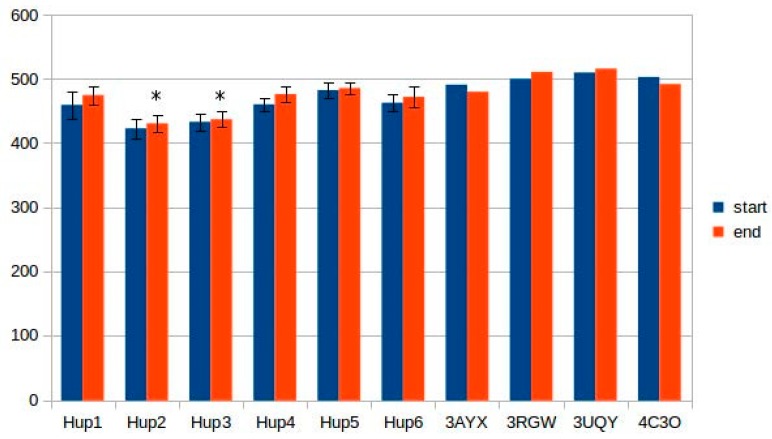
Number of hydrophobic contacts in the large subunits of HupSL hydrogenases. **Hup1**-HupSL from *Tca. roseopersicina*; **Hup2**-HupSL from *Rba. capsulatus*; **Hup3**-HupSL from *Rba. sphaeroides*; **Hup4**-HupSL from *Rps. palustris*; **Hup5**-HupSL from *Rvi. gelatinosus*; **Hup6**-HupSL from *Rsp. rubrum*; **3AYX**-HoxKG from *H. marinus*; **3RGW**-HoxKG from *R. eutropha*; **3UQY**-HyaAB from *E. coli*; **4C3O**-HydAB from *Sa. enterica.* Start-models (structures) before energy minimization; end-models (structures) after energy minimization. The data are represented as mean ± 2∙× SD. Significantly low results (ranges not overlapping with the others in the subgroup) are marked by asterisks.

**Figure 10 ijms-21-00366-f010:**
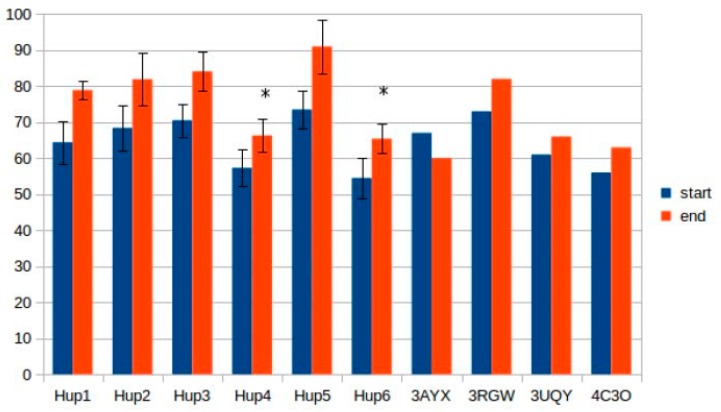
Number of ionic pairs in the large subunits of HupSL hydrogenases. **Hup1**-HupSL from *Tca. roseopersicina*; **Hup2**-HupSL from *Rba. capsulatus*; **Hup3**-HupSL from *Rba. sphaeroides*; **Hup4**-HupSL from *Rps. palustris*; **Hup5**-HupSL from *Rvi. gelatinosus*; **Hup6**-HupSL from *Rsp. rubrum*; **3AYX**-HoxKG from *H. marinus*; **3RGW**-HoxKG from *R. eutropha*; **3UQY**-HyaAB from *E. coli*; **4C3O**-HydAB from *S. enterica.* Start-models (structures) before energy minimization; end-models (structures) after energy minimization. The data are represented as mean ± 2 × SD. Significantly low results (ranges not overlapping with the others in the subgroup) are marked by asterisks.

**Figure 11 ijms-21-00366-f011:**
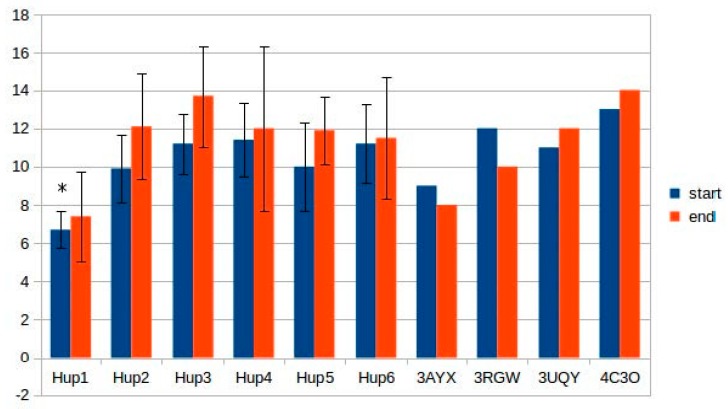
Number of ionic pairs between the subunits of HupSL hydrogenases. **Hup1**-HupSL from *Tca. roseopersicina*; **Hup2**-HupSL from *Rba. capsulatus*; **Hup3**-HupSL from *Rba. sphaeroides*; **Hup4**-HupSL from *Rps. palustris*; **Hup5**-HupSL from *Rvi. gelatinosus*; **Hup6**-HupSL from *Rsp. rubrum*; **3AYX**-HoxKG from *H. marinus*; **3RGW**-HoxKG from *R. eutropha*; **3UQY**-HyaAB from *E. coli*; **4C3O**-HydAB from *S. enterica.* Start-models (structures) before energy minimization; end-models (structures) after energy minimization. The data are represented as mean ± 2 × SD. Significantly low results (ranges not overlapping with the others in the subgroup) are marked by asterisks.

**Figure 12 ijms-21-00366-f012:**
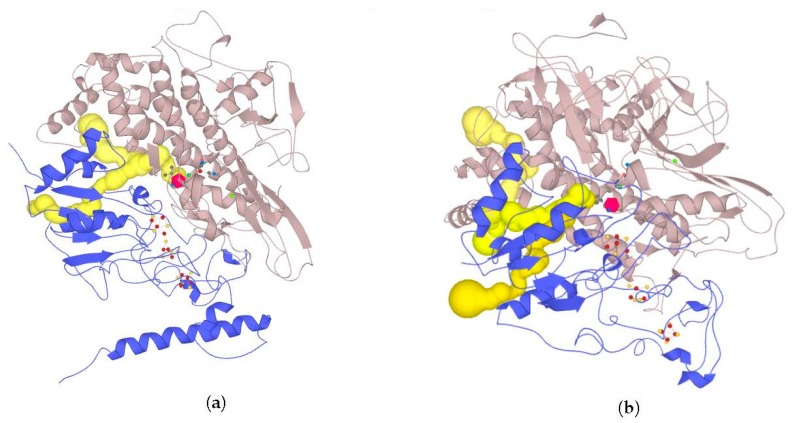
Tunnel structure in some of the studied hydrogenases. (**a**) HupSL from *Rsp. rubrum*; (data available online: https://mole.upol.cz/online/U1IDaUXAtk9G2dt1a54w/1); (**b**) HoxKG hydrogenase from *R/eutropha* (PDB ID: 3RGW); the most important gas channels (starting in the points close to active site) are colored yellow (data available online: https://mole.upol.cz/online/ORQHeeJ90OSUp1FWnBFlg/1); the red polyhedron indicates V77, one of the first lining residues of the tunnels close to the active site. The nickel atom is colored dark-green.

**Figure 13 ijms-21-00366-f013:**
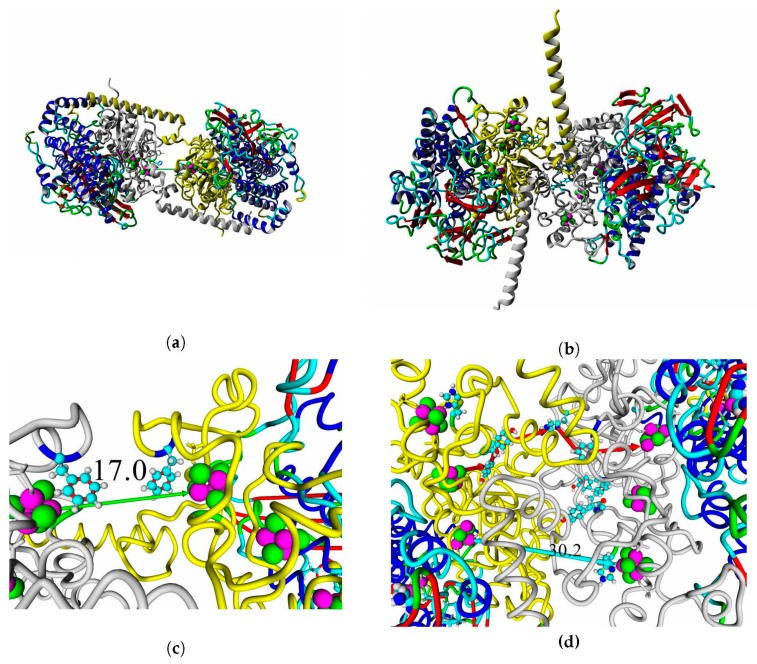
FeS cluster interaction in HupSL–HupSL hydrogenase oligomers obtained by protein–protein docking. Overall view of the oligomers: (**a**) HupSL from *Rba. capsulatus*; (**b**) HupSL from *Rba. sphaeroides*. The small subunits are colored gray and yellow. Close view of FeS clusters in the adjacent small subunits: (**c**) HupSL from *Rba. capsulatus*; (**d**) HupSL from *Rba. sphaeroides.* Aromatic residues are shown as balls and sticks colored by element. Green and cyan arrows illustrate the shortest paths of direct electron transfer (distance shown in Å); red arrows (**d**) show possible direction of electron “hopping” through the network of aromatic residues.

**Figure 14 ijms-21-00366-f014:**
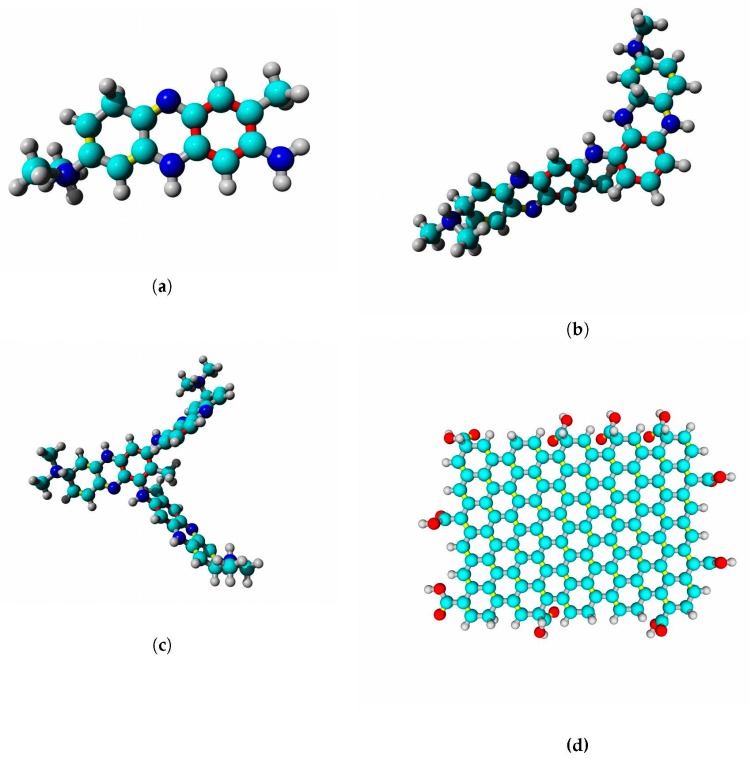
Substances used in molecular docking: (**a**) neutral red monomer; (**b**) neutral red dimer; (**c**) neutral red trimer; and (**d**) graphene oxide. Carbon is colored cyan, nitrogen is blue, oxygen is red, and hydrogen is gray. Single bonds are gray, double bonds are yellow, and resonance bonds (order 1.5) are red.

**Figure 15 ijms-21-00366-f015:**
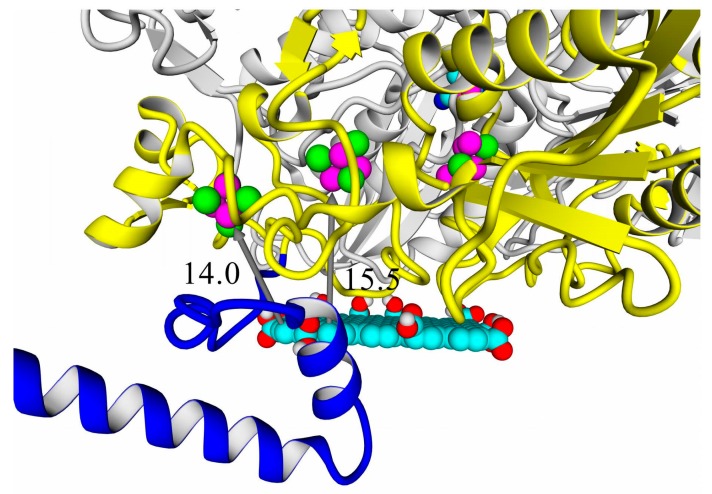
Docking complexes of *Rba. sphaeroides* HupSL hydrogenase and graphene oxide: productive complex. Distances from the aromatic plane of graphene oxide are denoted by gray arrows and labeled (14.0 Å from the plane to the nearest Fe atom in the distal FeS cluster; 15.5 Å from the plane to the medial FeS cluster). Coloring scheme: large subunit, gray; small subunit, main part, yellow; small subunit, C-end, blue; ligands and graphene oxide are colored by standard YASARA element scheme (carbon—cyan, oxygen—red, hydrogen—gray, sulfur—green, nickel and iron—magenta).

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
