# Peer review of "Hup-Type Hydrogenases of Purple Bacteria: Homology Modeling and Computational Assessment of Biotechnological Potential"

_ijms, 2020, doi:10.3390/ijms21010366_

Round 1

Reviewer 1 Report

The manuscript investigates the structures of six hydrogenases that have not been crystallized or extensively studied as potential targets for the biotechnological production of hydrogen.

The design of the study is sound and the authors performs the in-silico investigation keeping in mind the potential pitfalls of his approaches and searches for specific structural details that are highly relevant for a biotechnological use of the proteins.

While the study is generally well-written and the insights coming from the simulation are properly discussed, there are a few points that would improve the text, in my opinion.

General considerations:

1) The choice of dividing the description of the six target proteins between the introduction and the discussion is odd and makes the text harder to follow. I believe that moving all descriptions of the target proteins from the introduction and discussion sections to a dedicated paragraph at the beginning of the result section would be more appropriate, increasing the readability of the whole text.

2) While the author analyzes in detail his results in terms of the different protein subunits, function/folding of the C-termini, and potential dimer formation, there is no general description of the number/function/cofactors of subunits or of the general  tertiary/quaternary structure adopted by these hydrogenases. I believe that a paragraph defining these aspects (possibly an accompanying scheme or figure) would help the reader to better understand the choice of the results on which the text focuses.

3) The choice of neutral red and graphene oxide as small molecular targets as models of the the interface of an electrode is never justified (no reference is given in the introduction, for example).

4) The paragraph (line 602, page 20) suggesting that 'the number ionic interactions could presumably be related to proton transfer parameters' seems completely based off speculation. This observation should be better discussed, some references added or it could be skipped entirely. 

5) sometimes the 'good' results are found in the pre-optimized models, sometimes in the optimized ones. This is justified in the case of gas-tunnels, but not in other cases, for graphene oxide binding, for example. A rationalization should be given, if possible.

Specific corrections:

6) the author uses 'conservative' several times (for example line 152). Judging from the context he probably meant 'conserved'

7)line 61, page 2: the sentence is not clear, I suggest dividing it in two to make it clearer who is forming the dimer of heterodimers and where the electrons are coming from.

8)lines 285-286, page 10: '[...] transmembrane alpha-helix higher than outside or inside part of the protein;' This sentence should be rewritten, I cannot understand what it means.

9)line 543, 'but' should be omitted.

Author Response

Dear Reviewer! Thank you very much for your strict, rigorous and thorough review! Please see the attachment with response to your comments!

Best regards, Author.

Reviewer 2 Report

Please refer to the attached file for details.

Author Response

Dear Reviewer! I greatly appreciate your time and effort! Please see the attached file with answers to your comments!

Best regards, Author.

This manuscript is a resubmission of an earlier submission. The following is a list of the peer review reports and author responses from that submission.

Round 1

Reviewer 1 Report

The manuscript by Abdullatypov describes the homology and partly de novo 3D structure modelling of purple bacteria Hup-type hydrogenases. According to the hydrogenase database HydDB, these hydrogenases belong to the group 1d hydrogenases and are thus close relatives of the O2-tolerant, experimentally well-characterized, hydrogenases from E. coli (Hyd-1), H. marinus, R. eutropha (MBH), A. aeolicus, S. enterica (Hyd-5).

First the overall structures are compared, followed by the analysis of the microenvironment around the specialized [4Fe3S] cluster which has been shown to be involved in O2 tolerance of homologous hydrogenases. Next, the C-terminal regions of the respective small subunits were modelled de novo. Here, the tendency to mediate oligomerization was assayed. Subsequently, the full-size models, including the C termini, were compared followed by a subunit interaction analysis. Finally, molecular tunnels serving for gas transport and electrode/electron mediator-hydrogenase interactions were modelled.

In principle the presented data might be interesting and important, however, unfortunately, the author failed to include the aims of the study so that the context of the presented analyses is completely lost. Furthermore, the data are mainly presented as almost raw data (lines 151-172 would be great as sequence logo; tables 5, 6 and 7 are not comprehensible at all)  and are thus not comprehensible for the general reader. Figures 2, 5 and 6 are almost useless, since it is almost impossible to see the necessary details which are hidden in the overall structures.

In particular:

L51-52: the 4Fe3S cluster has been suggested to deliver electrons to the NiFe active site for O2 reduction

59-62: It can easily be concluded from the additional cysteines coordinating the 4Fe3S cluster (Goris et al., Fritsch et al.) that the Hup-type hydrogenases analyzed here, belong to the group of O2 tolerant enzymes. Greening et al. have also nicely drawn the same conclusion from their large sequence alignment presented in the HydDB. (The corresponding references of the database need to be cited by the way.) Therefore, the author has to make clear which additional information can be gained from the modelling of the structures.

64: Here is missing: what has been done and what is the purpose of the analysis. This holds true also for all the following chapters.

73-74: As described above: the presence of 4Fe3S clusters was already evident from sequence alignments

93: In this chapter it is unclear what was the benefit of the modelling. It seems that the data is purely based on sequence alignments.

95-96: It has not been shown that the proximal cluster reduces O2.

97-108: Why only residues from the large subunit have been investigated? Parts of the shown sequences do not belong to the microenvironment. There are residues like a glutamate from the small subunit that seem much more important. The alignment misses numbers for the reader to find the residues.

133-143: Residues should be labeled in the alignment

147: HydDB reference is missing

244-245: The structure of the actinobacterial hydrogenase is obviously severely different from the Hup-type hydrogenases which bind to a membrane-integral cytochrome b. This comparison does not make very much sense. Furthermore, there is a structure available by Volbeda et al. including the C-terminal part of E. coli Hyd-1.

276-277: As outlined above, these hydrogenases cannot be compared in this way

344: The author fully ignored the existing experimental data on gas tunnels available. The gas tunnels have been assayed so far using xenon, krypton and O2. The modelled structures need at least be compared with the experimental data.

393-394: The catalytic bias of this type of hydrogenase has been analyzed experimentally and should be cited accordingly.

Author Response

Dear Reviewer 1! Thank you for very critical review of the manuscript. Since you made such comments, I revised my article and did not highlight the proximal FeS cluster too much, because there are more other results that could be of equal importance. In general, attempts were made to modify the data that were commented as “raw”, but they couldn’t be all revised keeping their information capacity and level of detail the same. Since some troubles occurred during compression of the figures which lead to inevitable loss of clarity of their clarity, they were made larger and more clear.

Point 1: In principle the presented data might be interesting and important, however, unfortunately, the author failed to include the aims of the study so that the context of the presented analyses is completely lost. 

Response 1: Dear Reviewer 1! Thank you for this comment. The goals of the study were included explicitly to the revised version of the manuscript; the accents were changed in the concept of the article.

Point 2: Furthermore, the data are mainly presented as almost raw data (lines 151-172 would be great as sequence logo; tables 5, 6 and 7 are not comprehensible at all)  and are thus not comprehensible for the general reader.

Response 2: Dear Reviewer 1! Thank you for your advice, I tried to make a sequence logo, but with such a large difference in some cases (191F, 16Y, 7L or 201I, 12V, 1T) the data on minor residues present in several (1-5) enzymes from the group wouldn’t be clear; as for tables 5, 6 and 7, from my humble point of view, they contain valuable information that should be presented in the main part of the article, because they consider statistically confirmed differences between the interaction numbers and (Table 7) docking affinity scores (energies) that should also be shown because of their significance when comparing the modelled hydrogenases with structures from PDB in terms of possible interaction with electrode surface.

Point 3: Figures 2, 5 and 6 are almost useless, since it is almost impossible to see the necessary details which are hidden in the overall structures.

Response 3: Dear Reviewer 1! Thank you for such a critical comment. The figures in the pdf version were compressed severely; in this regard, Figure 2 was enlarged and residues were labeled, also, two subunits were coloured differently; Figure 6 (Figure 7 in the revised version) was enlarged to make the FeS clusters visible. Figure 5 was re-drawn too, since the tunnels needed to be mapped with a higher level of strictness; some new results appeared from re-mapping.

Point 4: L51-52: the 4Fe3S cluster has been suggested to deliver electrons to the NiFe active site for O2 reduction

Response 4: Dear Reviewer 1! Thank you for this comment, I have corrected this sentence in the revised version of the manuscript.

Point 5: 59-62: It can easily be concluded from the additional cysteines coordinating the 4Fe3S cluster (Goris et al., Fritsch et al.) that the Hup-type hydrogenases analyzed here, belong to the group of O2 tolerant enzymes. Greening et al. have also nicely drawn the same conclusion from their large sequence alignment presented in the HydDB. (The corresponding references of the database need to be cited by the way.) Therefore, the author has to make clear which additional information can be gained from the modelling of the structures.

Response 5: Dear Reviewer 1! You are right that presence of 4Fe3S clusters can be concluded from the primary sequences of the small subunits; as for data by Greening et al., they worked on large subunit alignments, although reflecting the presence of 4Fe3S clusters in Group1d in the description of the entries of the database, but they did not work with the small subunits alignments. The benefits of homology modelling is the spatial confirmation of something deduced from primary sequence: it could be excessive in almost all cases of high homology, but sometimes spatial orientation could be affected by some bulky residues or gaps in alignments, so the modelling increases the degree of reliability.

To shift the accent from the cluster to more common problem of biotechnological application, a corresponding phrase was added to introduction, and goals of the work were added to this section.

Point 6: 64: Here is missing: what has been done and what is the purpose of the analysis. This holds true also for all the following chapters.

Response 6: Dear Reviewer 1! The goals of the study were written in the Introduction (thanks to your previous comment).

Point 7: 73-74: As described above: the presence of 4Fe3S clusters was already evident from sequence alignments

Response 7: Dear Reviewer 1! As I have already said, the presence of 4Fe3S clusters is really evident, but it was not confirmed spatially; moreover, to study microenvironments of the FeS clusters, geometric analysis requires homology modelling. The phrase about spatial orientation “(oriented similarly in space, not just in primary structure)” was added to emphasize the importance of homology modelling.

Point 8: 93: In this chapter it is unclear what was the benefit of the modelling. It seems that the data is purely based on sequence alignments.

Response 8: Dear Reviewer 1! Sequence alignments are good to find out possible differences even in the microenvironments of the FeS clusters, but they are not enough to show the exact geometrical differences and, for example, to rank the varying residues by their possible electrochemical importance.

Point 9: 95-96: It has not been shown that the proximal cluster reduces O2.

Response 9: Thank you very much, this sentence was corrected according to your comments

Point 10: 97-108: Why only residues from the large subunit have been investigated? Parts of the shown sequences do not belong to the microenvironment. There are residues like a glutamate from the small subunit that seem much more important. The alignment misses numbers for the reader to find the residues.

Response 10: Dear Reviewer 1! This comment is really very valuable, so I had to analyze the small subunits and to make comments on this in the revised version of the manuscript (in the beginning of the Chapter 2.2). Moreover, the alignment was numbered properly.

Point 11: 133-143: Residues should be labeled in the alignment

Response 11: Dear Reviewer 1! The residues were labeled and highlighted in the alignment in the revised version

Point 12: 147: HydDB reference is missing

Response 12: Thank you very much, the reference was inserted into the final version of the manuscript

Point 13: 244-245: The structure of the actinobacterial hydrogenase is obviously severely different from the Hup-type hydrogenases which bind to a membrane-integral cytochrome b. This comparison does not make very much sense. Furthermore, there is a structure available by Volbeda et al. including the C-terminal part of E. coli Hyd-1.

Response 13: Dear Reviewer 1! The structure of the actinobacterial-type hydrogenase from R. eutropha is definitely very distant from the studied Hup-hydrogenases when considering the level of sequence homology; however, it is also a hydrogenase having C-terminal extension; moreover, it is the only hydrogenase where the direct interaction of C-terminal helices was shown to participate in oligomerization; as for the structure by Volbeda et al. (6G94), it shows more complex oligomerization: there is side-by-side interaction of the main parts of small subunits, there is end-to-end interaction of main parts of the small subunits with the outer parts of transmembrane cytochrome, and side-to-side interaction of alpha-helices of the extension of small subunits with alpha-helices of cytochrome; no direct oligomerization via interactions of the C-terminal extensions was shown. The idea to assess the involvement of C-terminal extensions into oligomerization was driven by works on Thiocapsa roseopersicina HydSL-hydrogenase by Sherman and colleagues (1991) showing the hexameric state; however, since no positive results were shown, and taking into account your comments, the part assessing the possibility of oligomerization via C-terminal extensions was removed from the revised version of the manuscript.

Point 14: 276-277: As outlined above, these hydrogenases cannot be compared in this way

Response 14: Dear reviewer 1! I made a detailed response in the previous point.

Point 15: 344: The author fully ignored the existing experimental data on gas tunnels available. The gas tunnels have been assayed so far using xenon, krypton and O2. The modelled structures need at least be compared with the experimental data.

Response 15: Dear Reviewer 1! Thank you for this comment. I found the data on krypton derivatization of tunnels in Ralstonia eutropha hydrogenase (PDB ID: 5D51) and oxygen derivatization in the same enzyme (PDB ID: 5MDL) and compared them with the data of tunnel mapping in the same hydrogenase (PDB ID: 3RGW). As a result, a new table was added to the manuscript comparing mapping in MOLE2.5 and in experiments of Kalms et al. The results were compared with mapping in R. rubrum hydrogenase taken for example.

Point 16: 393-394: The catalytic bias of this type of hydrogenase has been analyzed experimentally and should be cited accordingly.

Response 16: Dear Reviewer 1! Thank you very much, the catalytic bias was mentioned in the revised version of the manuscript and cited appropriately.

Reviewer 2 Report

The study by Abdullatypov entitled ‚Hup-type hydrogenases of purple bacteria: homology modeling reveals evidence for existence of oxygen-tolerant cluster’ is a theoretical approach to classify hydrogenases with regard to their oxygen tolerance based on sequence analysis, structure predictions and presence of a particular FeS-cluster that is commonly associated with oxygen tolerant hydrogenases.

The idea of such a study would be to discover and classify novel enzymes, but due to the theoretical nature of the study, the author presents experimental results from accompanying studies and thus the work presented here reads like a literature accumulation and lacks from practical experimental proof. Also no new oxygen tolerant hydrogenases were ‘discovered’ with this approach and personally I’d start by presenting the model process and then go further into detail with the location and properties of the FeS cluster prediction.

More specific comments:

Line 19: The two groups based on the numbers of interactions are not clearly defined. It is not clear what these have as consequences.

Line 52: What is meant by an adjacent hydrogenase ‘molecule’? The introduction lacks closer explanation of the architecture of these hydrogenase subunits. Therefore, it is never clear if the author refers to motifs within the large or small subunit, where the respective cofactor resides. Hence here the molecule could mean the other large subunit or the corresponding small subunit.

Lines 56/419: H2 – lower case for 2

Line 57: What is ‘absolute’ oxygen tolerance? The community has clear descriptions of the spectroscopic Ni-states that are involved.

Line 61 and others: It is very uncommon to write in first person.

Line 71: It is not clear what models and templates are here. No brief description of the modelling process is given and hence it is not clear what z-scores and their use is here.

Line 74: Again, this sounds as if the respective FeS cluster is located in the catalytic subunit. Were they modelled together? Were they already docked at this point? How do you see the cluster co-ordination here?

Line 78: Which is the model? Colours could be changed to more common ones (Fe brown, S – yellow) and the figure made clearer.

Line 80: ‘sulpur’

Line 85 (and others): Leave a space after the abbreviated first letter of the organism

Equation 1: this might be correct, but not an equation. Just state in the text that 2 Cys can replace one Si (and also mention what Si could stand for because commonly S2- is used). Please also explain with a little more detail why this cluster needs 2 extra Cys-residues.

Line 95: The FeS cluster does not reduce O2 to H2O, please also see the comprehensive review by Fritsch J, Lenz O, Friedrich B. Structure, function and biosynthesis of Oâ‚‚-tolerant hydrogenases. Nat Rev Microbiol2013;11:106–14.

Line 99: What part of the large subunit is shown? Which amino acids? What is their function or location in the structure?

Line 108: This would be HydB

Line 113: What was the proposed function of these residues (cluster binding, which subunit)?

Line 126: Models or structures? The figure is very crowded, that helix in fron covers most of the important residues.

Line 147: HydDB is mainly based on large subunits. Are these sequences in the large subunit?

Line 151: The consensus ‘figure’ is not self-explanatory. Please graph or use ‘sequence logo’

Line 194: Please make this a ‘Figure’, also not clear what the importance of the C-terminal anchor helix is for oxygen tolerance. To my knowledge no connection is known and this is not significant for the current study.

Lines 213/218/243/434: italics for E. coli, R. gelatinosus, R. eutropha…

Line 248: I cannot see an alignment in my pdf file

Line 267: I’d rather see the discussion of the docking of the models than from the published structures.

Line 286: Which enzyme models in particular. What influences this score?

Line 304: Is it HupS or HupSL? Were dimers of dimers attempted?

Tables 5 and 6 have no obvious relevance to the oxygen tolerance for me. In my opinion, they should be moved to supplemental data. Are the interactions correlated, e.g. the lower hydrogen for Rhodobacter come with higher ionic ones? Are the effects significant on the protein level e.g. in direct comparison of catalytic properties of the hydrogenases?

Line 335: ‘statistically’

Line 346: After…?

Line 347: Are the tunnels restricted in diameter for the gas molecules as one theory states that oxygen tolerance is conferred upon a hydrogenase by the limitation of O2 access to the active site? What diameters are common and observed here and e.g. in a O2-sensitive hydrogenase model?

Line 349: Why should there be a tunnel at the FeS? The electron transport distance should be sufficient. The FeS cluster is not directly attacked by O2.

Line 404: Please remain at the structure-function level of the discussion

Line 418: It would be a great advantage to determine the O2-tolerance of the enzymes in a practical experiment. It is not stated here whether this study identified new enzymes, at least not according to the HydDB.

Line 516: Truncation should not be to the lase Cys-residue but to the protease processing site which is usually 4 amino acids C-terminally

Line 545: I’m trying to understand the DOPE score. It states here they are good below -1 and line 286 states the opposite?

Line 616: Why is there a second (identical?) conclusions section? Unfortunately, this is very representative for the overall quality of the study.

Author Response

Dear Reviewer 2! Thank you very much for your valuable critical comments, they helped me to improve the article significantly. Please, see below the point-by-point answers to your comments.

Point 1: The idea of such a study would be to discover and classify novel enzymes, but due to the theoretical nature of the study, the author presents experimental results from accompanying studies and thus the work presented here reads like a literature accumulation and lacks from practical experimental proof. Also no new oxygen tolerant hydrogenases were ‘discovered’ with this approach and personally I’d start by presenting the model process and then go further into detail with the location and properties of the FeS cluster prediction.

Response 1: Dear Reviewer 2! The idea of the study was to apply computational methods to hydrogenases of purple bacteria in order to analyze them in comparison with other hydrogenases of this group. All the results obtained in the present work, as well as Discussion section, are aiming to show to the Community that there are many hydrogenases that are probably underestimated in terms of their prospects to be applied in hydrogen technologies; moreover, the approach demonstrated in the work could serve to find novel oxygen-tolerant hydrogenases in new genomes or metagenomes and comparing them with the existing and well-studied MBHs by the following scheme: 1) Sequence alignment; 2) Homology modeling; 3) Tunnel mapping; 4) Docking of electrode compounds. However, the results did not absolutely correspond to the expectations (for example, tunnel mapping did not show very reliable differences, although pointing on very interesting ones,  and the process could be a bit complicated and flaking to suggest any robust predictions), but the overall scheme could be used to analyze novel enzymes.

As for interactions in the subunits and between the subunits, this work definitely shows the need for analysis of quite big data (6 enzymes, each sampled by 10 models before and 10 models after energy minimization) in order to get some statistically significant values.

Of course, it would definitely be good to find some experimental confirmations of the results of the work. For example, if docking was correct, it could be confirmed by higher current densities of electrodes based on the enzymes of Rx.gelatinosus and Rba.sphaeroides, but the amount of work to be done is quite large and it could take a couple of years to carry out a comprehensive comparative study of these enzymes. So it could be the subject of further study, and I hope that my article would drive attention of the researchers from different parts of world and promote some kind of competition between the laboratories studying these enzymes.

Point 2: Line 19: The two groups based on the numbers of interactions are not clearly defined. It is not clear what these have as consequences.

Response 2: Thank you for this comment. The abstract was re-written in order to make this phrase shorter and clearer; the detailed division into two groups is described furhter.

Point 3: Line 52: What is meant by an adjacent hydrogenase ‘molecule’? The introduction lacks closer explanation of the architecture of these hydrogenase subunits. Therefore, it is never clear if the author refers to motifs within the large or small subunit, where the respective cofactor resides. Hence here the molecule could mean the other large subunit or the corresponding small subunit.

Response 3: Dear Reviewer 2! The adjacent hydrogenase molecule means a second SL-heterodimer in a dimer of heterodimers; the sentence was corrected in the revised version of the manuscript

Point 4: Lines 56/419: H2 – lower case for 2

Response 4: Thank you very much, the mistakes were corrected in the revised version.

Point 5: Line 57: What is ‘absolute’ oxygen tolerance? The community has clear descriptions of the spectroscopic Ni-states that are involved.

Response 5: Dear Reviewer 2! Absolute oxygen tolerance was attributed to HupUV hydrogenases in the introduction based on the data by Duche et al., 2005: aerobiosis did not decrease the rate of H-D exchange, H-D formation and H2 formation by hydrogenase from Rhodobacter capsulatus. As far as I know, the HupUV hydrogenases are not inhibited by oxygen at all; the mechanism of their oxygen tolerance is attributed to the narrow gas channel formed by bulky residue sidechains; but for me it is quite strange why they are studied not as intensely as membrane-bound hydrogenases of E.coli, R.eutropha and other bacteria. The data in the literature are much less abundant than for MBH, and I do not know what spectroscopic Ni-states do determine their oxygen tolerance, in contrast to MBH or selenium-containing hydrogenases of sulfate-reducing bacteria.

Point 6: Line 61 and others: It is very uncommon to write in first person.

Response 6: Thank you, all the personal pronouns were changed to “the author”

Point 7: Line 71: It is not clear what models and templates are here. No brief description of the modelling process is given and hence it is not clear what z-scores and their use is here.

Response 7: Dear Reviewer 2! The sentence was changed to clear things out in the revised version of the article.

Point 8: Line 74: Again, this sounds as if the respective FeS cluster is located in the catalytic subunit. Were they modelled together? Were they already docked at this point? How do you see the cluster co-ordination here?

Response 8: Dear Reviewer 2! Two subunits were modelled simultaneously and all the ligands were docked into the models of respective subunits (advanced MODELLER protocols allow doing that).

Point 9: Line 78: Which is the model? Colours could be changed to more common ones (Fe brown, S – yellow) and the figure made clearer.

Response 9: Dear Reviewer 2! “Model” and “X-ray structure” were shown explicitly in the figure caption. As for the colouring scheme, it is the common colouring from YASARA View.

Point 10: Line 80: ‘sulpur’

Response 10: Thank you, the mistake was corrected.

Point 11: Line 85 (and others): Leave a space after the abbreviated first letter of the organism

Response 11: Thank you very much, the mistake was corrected in the revised version.

Point 12: Equation 1: this might be correct, but not an equation. Just state in the text that 2 Cys can replace one Si (and also mention what Si could stand for because commonly S2- is used). Please also explain with a little more detail why this cluster needs 2 extra Cys-residues.

Response 12: Thank you for this correction, it really helped me to formulate the principle of reduction of required S number more accurately. The equation was deleted from the revised version, and it was explained in plain text.

Point 13: Line 95: The FeS cluster does not reduce O2 to H2O, please also see the comprehensive review by Fritsch J, Lenz O, Friedrich B. Structure, function and biosynthesis of Oâ‚‚-tolerant hydrogenases. Nat Rev Microbiol2013;11:106–14.

Response 13: Thank you very much for your comment, the sentence was corrected to provide more exact description of action of the cluster.

Point 14: Line 99: What part of the large subunit is shown? Which amino acids? What is their function or location in the structure?

Response 14: Dear Reviewer 2! This part of the large subunit is close to the proximal FeS cluster, and the residues E73 and H229 were shown to be determinants of oxygen stability (reversibility of inhibition by oxygen) by Bowman and colleagues, as it is discussed further. Other residues are just close to the mentioned residues; moreover, they differ in some of the studied hydrogenases, and they could affect some electrochemical properties due to their difference in charge/polarity.

Point 15: Line 108: This would be HydB

Response 15: Dear Reviewer 2! Initially I named this enzyme HyaAB according to its Uniprot name; however, I must admit that you should be definitely more competent in hydrogenase nomenclature than me, so I changed the enzyme’s name to HydAB following your comment

Point 16: Line 113: What was the proposed function of these residues (cluster binding, which subunit)?

Response 16: Dear Reviewer 2! The proposed function of E73 and H229 (large subunit) is fine tuning of the electrochemistry of FeS cluster. For example, H229 was shown to be a determinant of overpotential requirement (along with oxygen stability); E73 was supposed to be coordinating H229; as for two other residues, V/T and F/Y, they could also modulate specific electrochemical properties due to their difference in polarity, as was supposed in the following paragraphs

Point 17: Line 126: Models or structures? The figure is very crowded, that helix in fron covers most of the important residues.

Response 17: Dear Reviewer 2! These are models superposed onto each other; the figure was made clearer, and it was also enlarge in order to make it more visible.

Point 18: Line 147: HydDB is mainly based on large subunits. Are these sequences in the large subunit?

Response 18: Dear Reviewer 2! These sequences are in the large subunit, and it was stated more clear in the revised version

Point 19: Line 151: The consensus ‘figure’ is not self-explanatory. Please graph or use ‘sequence logo’

Response 19: Dear Reviewer 2! Thank you for this comment. Of course, graph or sequence logo could reflect most of the differences in the structure of these parts of large subunits in quite a beautiful way, but in cases when only minor portion of sequences (1-5 of 214) possessed certain residues, my sequence logos lost their clearance and so I had to keep this part of the article without changes.

Point 20: Line 194: Please make this a ‘Figure’, also not clear what the importance of the C-terminal anchor helix is for oxygen tolerance. To my knowledge no connection is known and this is not significant for the current study.

Response 20: Dear Reviewer 2! The alignment was transformed into a figure; as I have already mentioned, the Title and Introduction were revised so that the reader’s attention wouldn’t focus too much on the oxygen tolerance. Moreover, the membrane embedding of the C-terminal extension could take part in formation of oxygen-tolerant hydrogen-oxidizing complex (hydrogenase-cytochrome) in vivo, but I cannot assess the real contribution of membrane anchoring into this complicated process quantitatively.

Point 21: Lines 213/218/243/434: italics for E. coli, R. gelatinosus, R. eutropha…

Response 21: Thank you for this comment, this mistake was corrected in the revised version.

Point 22: Line 248: I cannot see an alignment in my pdf file

Response 22: Dear Reviewer 2! The alignment was removed from the main part of the manuscript in the revised version.

Point 23: Line 267: I’d rather see the discussion of the docking of the models than from the published structures.

Response 23: Dear Reviewer 2! Protein-protein docking could be really fruitful, but as far as I know, it is quite complicated, especially given the fact that hydrogenases contain ligands that could not be recognized by many docking servers, and their removal could affect the scoring of docking results; when considering only C-terminal parts of the small subunits, it could be simpler; however, if I would start docking the mentioned C-termini, I would like to combine it with either Monte-Carlo or molecular dynamics to verify the results and make them more reliable. This could be a subject of a separate study.

Point 24: Line 286: Which enzyme models in particular. What influences this score?

Response 24: Dear Reviewer 2! The answer on your question lies in the Table 4 after this paragraph. This score is influenced by presence of C-terminal extension because of geometrical reason, making a protein with extension very different from ordinary globular protein, thus affecting the pairwise interatomic distance distribution, a parameter on which the DOPE statistical potential was based.

Point 25: Line 304: Is it HupS or HupSL? Were dimers of dimers attempted?

Response 25: Dear Reviewer 2! I am sorry for this mistake that could confuse the reader and thank you for this comment. It is HupSL, and the corrections were made in the legend in the revised version.

Point 26: Tables 5 and 6 have no obvious relevance to the oxygen tolerance for me. In my opinion, they should be moved to supplemental data. Are the interactions correlated, e.g. the lower hydrogen for Rhodobacter come with higher ionic ones? Are the effects significant on the protein level e.g. in direct comparison of catalytic properties of the hydrogenases?

Response 26: Dear Reviewer 2! You are right, these interactions do not have relevance to possible oxygen tolerance; however, they could reflect some possible differences in medium-dependent catalytic properties (stability of catalyst under high salt or at high temperatures), which require experimental verification. The experimental data confirming the supposal deduced from the analysis of ionic pairs for Rhodospirillum rubrum hydrogenase could be its stability to high salt concentration, which is stated in Discussion. However, since the article was revised according to the Reviewers’ comments, the focus of reader’s attention was partially shifted from oxygen tolerance to other prospects of biotechnological application of these enzymes, which was reflected in the Title and Introduction. Since these tables show statistically reliable data, I would like to keep them in the main part of the Manuscript.

Point 27: Line 335: ‘statistically’

Response 27: Thank you, the mistake was corrected.

Point 28: Line 346: After…?

Response 28: Thank you, this word was removed from the revised version.

Point 29: Line 347: Are the tunnels restricted in diameter for the gas molecules as one theory states that oxygen tolerance is conferred upon a hydrogenase by the limitation of O2 access to the active site? What diameters are common and observed here and e.g. in a O2-sensitive hydrogenase model?

Response 29: Dear Reviewer 1! The automatic settings of MOLE2.0 server limit the radius of tunnel by 1.2 Å, which is in good agreement with doubled atomic radius of oxygen (1.2 Å) – in other words, the diameter of tunnel is enough for oxygen molecule to pass; also, there is a parameter called “bottleneck length”: it is maximal length of tunnel part with a diameter less than 1.2 Å cut-off (3 Å by default; shifted to 0 Å to make computation stricter). Usually, the bottlenecks are not less than 0.7 Å. Following your comment, I checked the O2-sensitive hydrogenase from Desulfovibrio (PDB ID: 1H2A), and found that the tunnels leading to the active site are comparable, although sometimes slightly wider in the oxygen-sensitive hydrogenase. However, I wouldn’t like to make some statements without really thorough and time-consuming analysis. Meanwhile, it is a question what radii to use – either molecular (1.2 Å) or kinetic (1.73 Å), which could be definitely more appropriate in case of modeling the movement of a number of molecules in a large tunnel, but the diffusion of oxygen in hydrogenase molecule looks more like movement of a single molecule. I’ve just submitted tunnel computations with both parameters for oxygen-tolerant (3RGW) and oxygen-sensitive (1H2A) hydrogenases and I cannot say that I see some definitely evident differences deserving the reader’s attention (1.73 Å cutoff did not lead to mapping of any tunnels close to the active site, whereas 1.2 Å showed basically the same scheme of tunnels). The reasons for tunnel mapping aimed to find differences inside the group of oxygen-tolerant hydrogenases,because even they are inhibited by oxygen, though reversibly. Up to date, when I computed tunnels in a stricter condition than before (zero bottleneck tolerance), I found some differences: for example, tunnel networks in hydrogenases from Hydrogenovibrio marinus, Thiocapsa roseopersicina, Rhodobacter capsulatus, Rhodobacter sphaeroides look unfavorable for oxygen diffusion compared to other studied hydrogenases; but, dealing with such a complicated process, I ought to look at my own results very critically and to try not to overestimate the reliability of my results.

Point 30: Line 349: Why should there be a tunnel at the FeS? The electron transport distance should be sufficient. The FeS cluster is not directly attacked by O2.

Response 30: Dear Reviewer 2! Your comment is fair; however, there can be a lot of technical problems in the selection of starting point, so the reason to fix the point between NiFe center and FeS cluster was an attempt to calculate the tunnels from the point somehow close to the active site; centering the origin of tunnels on the active site alone did not lead to positive results. In fact, even centering of the origin on NiFe and FeS did not lead to solid results, so automatic starting point selection was the only approach giving the results that could be linked to the possible oxygen attack on the active site.

Point 31: Line 404: Please remain at the structure-function level of the discussion

Response 31: Dear Reviewer 2! The goal of writing this paragraph was to gather experimental information available for this enzyme and then (regarding structure-function level of discussion) to link it to some of the results of modeling (such as low level of ionic pairs in the large subunit, that could explain its high tolerance to salt).

Point 32: Line 418: It would be a great advantage to determine the O2-tolerance of the enzymes in a practical experiment. It is not stated here whether this study identified new enzymes, at least not according to the HydDB.

Response 32: Dear Reviewer 2! Thank you for this comment! This study did not describe absolutely new enzymes. All the modeled enzymes except hydrogenase from Rubrivivax gelatinosus, were studied experimentally by other research teams, but the designs of the studies were insufficient to regard these enzymes as “oxygen-tolerant” or “oxygen-sensitive”, because they were not studied like, for example, hydrogenase from S. enterica in the work by Lisa Bowman and colleagues. At least the present study shows high biotechnological potential of these hydrogenases comparable to that of the enzymes from E. coli, R. eutropha, S. enterica, Hv. marinus. As for correct assessment of oxygen tolerance, it requires very accurate electrochemical measurement and I hope that I could do it in further studies.

Point 33: Line 516: Truncation should not be to the lase Cys-residue but to the protease processing site which is usually 4 amino acids C-terminally

Response 33: Dear Reviewer 2! You are absolutely right, and to clarify this, the header of the Table has been rewritten in the revised version of the manuscript.

Point 34: Line 545: I’m trying to understand the DOPE score. It states here they are good below -1 and line 286 states the opposite?

Response 34: Dear Reviewer 2! The models are very good if their normalized DOPE z-score is below -1. But since DOPE statistical potential was calibrated on a set of soluble globular proteins, and the C-termini of the small subunits are extended alpha-helices, the values for full-sized models of the small subunits were above -1. Since the ab initio predicted C-terminal part is a priori a zone of less reliability because of lack of homologue with known 3D-structure, and ranking of models by quality should be more reliable in context of molecular interaction assessment (which involves only the main globule and does not involve the C-terminal helical part), the aligned parts of the small subunits were assessed to select 10 best models for each enzyme and to calculate the interaction numbers.

Point 35: Line 616: Why is there a second (identical?) conclusions section? Unfortunately, this is very representative for the overall quality of the study.

Response 35: Thank you very much for this comment, the duplicated conclusions section was deleted from the revised version of the manuscript.

Round 2

Reviewer 1 Report

The author has improved the manuscript to some degree.

However, Figure 2 still is not presented in a comprehensible manner. The amino acid numbering is not readable. I suggest to disable shadows and set the much less important cartoon representation to partial transparence. A major part in the right hand side part of the figure shows unimportant parts of the protein.

Line 138: Aquifex aeolicus should be included in the alignment since it is mentioned in line 231.

Page 7 : I am still confident that the general reader is not willing to read through raw data. The author says " the data on minor residues present in several (1-5) enzymes from the group wouldn’t be clear" and this is of course the meaning of a Logoplot, to show which residues are statistically relevant and which ones are not. Alternatively, I could imagine a figure with bargraphs or somthing similar and even a logarithmic scale can be used to point out underrepresented amino acids. However, it is not the job of the reviewer to convince the author of an intelligible data representation.

Fig. 3: Having a look into the structure file 4GD3 I noticed that the two small subunits show different helical regions in the C-terminal ends. In the second helix the helical region ranges from TADT...AVGVAHV fitting better the TMHMM prediction. This should be taken into account.

Tables 4/5/6/7/8 still need to be turned into human readable data.

Line 616ff: The findings of electrode material interaction should not only be compared to the mentioned Thiocapsa data but should also be compared with the findings from the following study: https://doi.org/10.1371/journal.pone.0143101

Reviewer 2 Report

This revised version of a study conducted by Abdullatypov has slightly improved in clarity e.g. by introducing the aims of the experimental design. However, the manuscript has not been restructured, the figures are still very difficult to understand and the line of thought remains difficult to follow. The author is very focussed on discussing the details of the modelled structures, but lacks to confer their meaning to the reader. Some of the answers that were given in response to the reviewers comments would also be helpful within the document for the reader. There is no gain in knowledge when e.g. predicting the transmembrane helices or a FeS-microenvironment that displaces the co-ordinating ligands when at the same time practical evidence for the suggested differences is missing. Overall, this study lacks novel findings (or presentation thereof) and is a proof-of-concept bioinformatical study.